# Towards Generalizable Retina Vessel Segmentation with Deformable Graph Priors

**Ke Liu** [1]    **Shangde Gao** [1]    **Yichao Fu** [1]    **Shangqi Gao** [2*]

[1] Zhejiang University
[2] University of Cambridge
{kliu, gaosd, fuyichao}@zju.edu.cn
sg2162@cam.ac.uk

## Abstract

Retinal vessel segmentation is critical for medical diagnosis, yet existing models often struggle to generalize across domains due to appearance variability, limited annotations, and complex vascular morphology. We propose GraphSeg, a variational Bayesian framework that integrates anatomical graph priors with structure-aware image decomposition to enhance cross-domain segmentation. GraphSeg factorizes retinal images into structure-preserved and structure-degraded components, enabling domain-invariant representation. A deformable graph prior, derived from a statistical retinal atlas, is incorporated via a differentiable alignment and guided by an unsupervised energy function. Experiments on three public benchmarks (CHASE, DRIVE, HRF) show that GraphSeg consistently outperforms existing methods under domain shifts. These results highlight the importance of jointly modeling anatomical topology and image structure for robust generalizable vessel segmentation. Code can be found at github.com/AI4MOL/GraphSeg.

## 1  Introduction

Retinal vessel segmentation is vital to assist in the diagnosis of common retinal diseases, such as diabetic retinopathy, age-related macular degeneration, and retinal detachment, which have been recognized as the leading causes of vision impairment and blindness globally by the World Health Organization (WHO) [2]. However, automatic segmentation is challenging due to high image heterogeneity. Particularly, retinal structures are changing with age, and some structures could be overlapped or occluded due to retinal lesions [1]. Besides, imaging artifacts widely exist in retinal images due to low contrast, eye movement, and noise, increasing the difficulties of distinguishing fine structures like blood capillaries [2].

Conventional supervised models have shown promising performance in automatically recognizing vascular structures and segmenting retinal vessels [3–5]. However, these models expect manual annotation of retinal vessels, which requires expert ophthalmologists and is labor-intensive, especially for blood capillaries [6]. Besides, while automatic vessel segmentation models have shown promising performance in recognizing vascular structures, they struggle to distinguish fine structures mixed with lesions or imaging artifacts, leading to discontinuous segmentation that breaks vascular structures. Moreover, supervised models tend to be over-dependent on training data when annotations are limited, and thus often deliver poor generalizability in unseen scenarios [7]. Furthermore, although large amounts of unannotated images are readily available, models trained solely with supervised losses often fail to generalize to unseen domains, due to their inherent reliance on labeled data.

---

*Corresponding author.
[2] https://www.who.int/news-room/fact-sheets/detail/blindness-and-visual-impairment

39th Conference on Neural Information Processing Systems (NeurIPS 2025).

Recent Bayesian approaches have shown promising generalizability by imposing statistical priors on medical image segmentation [8, 9]. These methods built probabilistic graphical models to describe the statistical correlation among images, noises, and targets. By statistically modeling local neighboring systems of pixels, they showed impressive ability in capturing compact shapes of organs, such as the heart and prostate. However, they overlooked the morphological characteristics of vascular structures such as retinal arteries, veins, and capillaries. Compared with compact shapes, vascular shapes have a higher surface-to-volume ratio, and the local modeling approaches lacked sensitivity in capturing long-range dependencies between pixels along vascular structures. Few works have focused on learning the long-range morphology of vessels by fitting skeleton priors extracted from ground truth [4], but they lack adaptability for unseen domains. Graphs have proven effective for representing non-Euclidean or irregular structures. Recent advances in graph shape analysis showed that retinal vessel structures can be effectively represented by principal graph components, independent of a specific image and its annotation [10, 11]. Hence, it is promising to incorporate image-agnostic retinal graph priors into vessel segmentation for better model generalization.

Motivated by the success of Bayesian image segmentation and graph shape analysis, we propose a Bayesian retinal vessel segmentation framework for modeling vascular shapes by imposing retinal graph priors. Specifically, we propose a probabilistic graphical model by jointly modeling images, vascular shapes, and the retinal graph atlas. To mitigate the spatial misalignment structure between vascular shapes and the retinal graph atlas, we introduce a deformable retinal graph prior from the atlas based on the learnable displacement field between vascular shapes of images and graph priors. To drive structural matching between images and priors, we develop a statistical model for the vascular shape, aiming to maximize the geometric similarity between vascular shapes and deformable graph priors. Based on variational inference, we build deep neural networks to solve the probabilistic graphical model and achieve promising generalizability in unseen scenarios, demonstrating the effectiveness of incorporating retinal graph priors in vessel segmentation.

As a summary, our main contributions are: (1) We propose a Bayesian retinal vessel segmentation framework for modeling vascular structures, which overcomes the limitations of conventional Bayesian image segmentation models only focusing on compact shapes; (2) We develop deformable retinal graph priors for matching heterogeneous vessel structures in retinal images, and design a statistical model for vascular structures to drive the spatial alignment between image vascular structures and deformable graph priors; and (3) We validate the Bayesian retinal vessel segmentation framework on multiple commonly used datasets, and demonstrate its superior generalizability for unseen scenarios.

## 2  Related Works

### 2.1  Retinal Vessel Segmentation

Retinal vessel segmentation has emerged as a critical tool in healthcare, providing non-invasive biomarkers for cardiovascular risk assessment [1, 2]. Although deep learning methods, including U-Net variants (*e.g.*, ResU-Net [12], FR-Net [5]), and Transformer-based hybrid frameworks [13, 14], with their symmetric encoder-decoder structure and skip connection [3], have significantly improved segmentation accuracy, several persistent challenges hinder clinical adoption. Specifically, disconnected vessel predictions and spurious bifurcations can compromise the reliability of diagnostic procedures [2]. Moreover, the model's robustness is limited by factors such as performance degradation under low-contrast conditions, the presence of imaging artifacts, and variability across different devices. Existing graph-based methods [15, 11] offer a partial solution to these issues, which are promising for representing non-Euclidean or irregular structures. However, they lack explicit anatomical constraints, resulting in implausible structures during domain shifts. While Generative Adversarial Networks (GANs) [16] enable the segmentation and reconstruction of blood vessel networks with no human input, they introduce instability during the training process, a problem especially prevalent in pathological cases. To address these issues, we propose a variational Bayesian framework, integrating deformable graph priors with a displacement field, which offers a principled and effective path toward anatomically consistent and generalizable medical image segmentation.

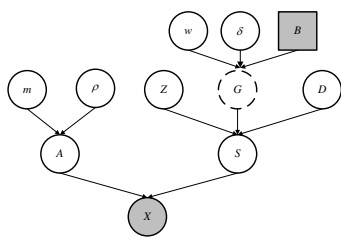

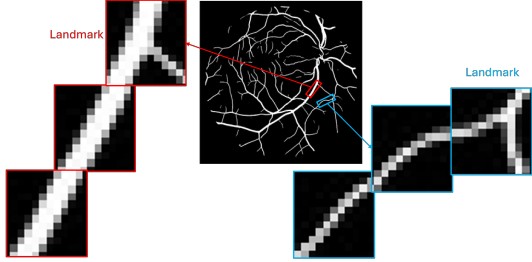

(a) Probabilistic Graphical Model.  (b) Geometric similarity and dissimilarity

Figure 1: Overall framework. (a) shows the probabilistic graphical model. White circles denote variables, and dashed white circles denote prunable variables. Specifically, $X$ denotes the observed image, $A$ and $S$ are structure-degraded and structure-preserved components respectively, $Z$ denotes the probability of foreground, and $G$ represents a deformable graph determined by linear weights $\mathbf{w}$, displacement field $\delta$, and a set of principal graph components $B$ (agnostic to $X$). (b) shows the geometric similarity between adjacent vascular segments and the dissimilarity between landmarks and its neighbors.

## 2.2 Image Decomposition

Image decomposition is widely explored, driven by the assumption that underlying data often lies in a low-dimensional subspace [17, 18, 9], offering strong potential in medical image analysis. Principal component analysis (PCA) demonstrates that the low-rank components of matrices are its basis elements [18]. Babacan et al. [17] introduced variational Bayesian methods for posterior inference, though their practical adoption has been limited by computational complexity. Recently, deep learning-based image decomposition methods have emerged. Deep image prior (DIP) [19] is introduced to capture essential image statistics from a single observation, enhanced DIP for image denoising [20], and RONet for efficient subspace interpretable learning [21]. BayeSeg [22, 8] addresses the challenges of domain generalization for medical image segmentation, which decomposes radiological images into compact shape and appearance variables. The segmentation process is then modeled as a locally smooth variable dependent only on shape features through a variational Bayesian inference framework. These innovations highlight the Bayesian framework's unique capacity to integrate statistical prior knowledge for promising generalizability, a pivotal benefit for retinal vessel segmentation.

## 2.3 Domain Generalization for Medical Image Segmentation

Domain generalization (DG) aims to train models on one or multiple source domains that robustly generalize to unseen target domains [7]. Current DG approaches for vessel segmentation broadly divide into three categories: **Data-centric augmentation** methods enrich training diversity via synthetic domain shifts [23, 24]. AADG [23] optimizes data augmentation policies through Sinkhorn distance-based diversity maximization and reinforcement learning to enhance cross-domain generalization. **Meta-learning methods** optimize for generalization through episodic training [25, 26]. Liu et al. [25] address both centralized and federated learning scenarios through shape-aware meta-objectives and frequency-space interpolation to enhance model robustness against domain shifts. **Domain-invariant representation methods** focus on domain-agnostic feature extraction [27–29]. A Hessian-based vector field [28] is proposed to model vessel structures as domain-invariant features, achieving superior cross-domain generalization in retinal vessel segmentation. Despite the progress, retinal vascular segmentation faces significant domain shifts due to differences in imaging artifacts, and the morphological characteristics of vascular structures, and existing studies remain underexplored. In particular, the lack of transparency in feature invariance of adversarial and meta-learning methods hinders clinical trust. Domain alignment strategies often fail to maintain microvascular continuity at low contrast variations.

## 3 Methodology

In this section, we present our method for generalizable retinal vessel segmentation, which integrates the structural graph prior into segmentation networks through a variational Bayesian framework. As

illustrated in Fig. 1a, we first introduce the probabilistic graphical model (PGM) that captures the underlying structure of retinal vessels by matching with the deformable graph prior, followed by a detailed description of the variational inference process that enables us to learn the model parameters and perform segmentation. For convenience, *we denote the vectorization of $X$ by* $\mathbf{x}$, *consistent with the notation used elsewhere.*

## 3.1 Deformable Graph Prior Guided Decomposition and Segmentation

**Image Decomposition.** Artifacts in retinal images could lead to unsatisfactory segmentation deviated from vascular structures. Motivated by the success of image decomposition in disentangling shape information, we decompose an image into a structure-preserved component, $S$, and a structure-degraded counterpart, $A$, as shown in Fig. 1a. The former presents the enhanced vascular structure closer to the ground truth, whereas the latter contains noisy structures degraded by vessel-like artifacts that adversely affect segmentation. Concretely, let $X \in \mathbb{R}^{h \times w}$ denote a grayscale image, we decompose $X$ into a structure-preserved variable $S$ and a structure-degraded variable $A$, i.e., $X = S + A$. By assuming $A$ follows a Gaussian distribution with mean $\mathbf{m}$ and inverse variance $\rho$, the distribution $p(X|S, A)$ can be expressed as, $p(X|S, A) = \mathcal{N}(X|\mathbf{s} + \mathbf{m}, diag(\rho)^{-1})$. To increase the capacity of $A$ in modeling artifacts, $\mathbf{m}$ is assumed to be a variable follows a Gaussian prior $\mathcal{N}(\mathbf{m}|\mu_m^0, (\sigma_m^0)^{-1}I)$, and $\rho$ is assumed to be a variable follows a Gamma prior $\mathcal{G}(\rho|\phi_\rho^0, \gamma_\rho^0)$.

**Deformable Graph Prior.** Conventional Bayesian approaches directly used manual annotations to guide the learning of anatomical structures [8]. They presented promising generalizability in cross-domain segmentation, but are limited to compact anatomy with a low surface-to-volume ratio. To encourage models to accurately capture vascular structures, we propose a deformable graph prior to guide image decomposition and vessel segmentation. Specifically, let $\mathcal{A} = \{G_1, \ldots, G_N\}$ denote a graph atlas, where graphs $\{G_n\}_{n=1}^N$ have different vertex sets $\{V_n\}_{n=1}^N$, while sharing a consistent vascular structure defined by a common edge set $E$, and $B = \{B_1, \ldots, B_K\}$ denote $K$ principle graph components of $\mathcal{A}$. Furthermore, let $G = \mathbf{w}^T B + \delta$, where the multiplication and addition operations are applied only for the vertex set of $B$, then the parametric graph $G$ determines a deformable graph prior, as shown in Fig. 1a. Here, $\mathbf{w} \in \mathbb{R}^K$ follows a Gaussian prior $\mathcal{N}(\mathbf{w}|0, (\sigma_w^0)^{-1}I)$ and determines a linear combination. $\delta$ follows another Gaussian prior $\mathcal{N}(\delta|0, (\sigma_\delta^0)^{-1}I)$ and defines a displacement field.

**Landmark Detection.** In general, adjacent vascular segments exhibit high geometric similarity along the direction of vessels. However, this pattern is disrupted when vessels are branching points, which are referred to as landmarks, as shown in Fig. 1b. Detecting these landmarks is important since they disobey geometric similarity and reversely affect vascular structure matching between the structure-preserved component $S$ and the deformable graph prior $G$. To detect landmarks, we introduce a variable, $D$, in Fig. 1a which follows a Gamma prior, $\mathcal{G}(\mathbf{d}|\phi_d^0, \gamma_d^0)$.

**Vessel Segmentation.** Let $Y \in \mathbb{R}^{h \times w}$ denote a vessel annotation of $X$, and $Z \in \mathbb{R}^{h \times w}$ denote the probability of vessel segmentation, where $p(Z)$ follows a Beta prior, $\mathcal{B}(\mathbf{z}|\alpha_z^0, \beta_z^0)$. Then the distribution $p(Y|Z)$ determines a supervised loss function for segmentation, such as cross-entropy or Dice loss. However, such pixel-wise loss functions struggle to capture vascular structures effectively, as they overlook long-range dependencies between pixels along the vessels. Motivated by the advantage of retinal graphs in preserving vascular structures, we develop structure-preserved image decomposition and vessel segmentation guided by the deformable graph prior. Concretely, let $\mathcal{H}(S; Z, G, D)$ denote an energy function that quantifies the quality of vascular structure matching between $S$ and $(Z, G)$ based on detected landmarks $D$, then $p(S|Z, G, D) \propto \exp\{-\mathcal{H}(S; Z, G, D)\}$ determines a deformable graph prior-guided image decomposition and vessel segmentation. Nevertheless, the definition of $\mathcal{H}(S; Z, G, D)$ is nontrivial and will be detailed in the following section.

## 3.2 Statistical Modeling of Vascular Structure

**Implicit Neural Representation.** The first challenge of defining $\mathcal{H}(S; Z, G, D)$ lies in measuring the structural similarity between $S$ and $G$. However, $S$ is pixel-wise, while the vertex set of $G$ is low-dimensional and sparse, making it difficult to directly define the structural similarity between $S$ and $G$. To tackle the difficulty, we used implicit neural representation to map $G$ to the same space as $S$. Concretely, let $f_\zeta(\cdot)$ denote a graph neural network parameterized by $\zeta$ to predict the Gaussian parameters of each vertex, then we can sample a Gaussian cloud from $f_\zeta(G)$ by Gaussian splatting,

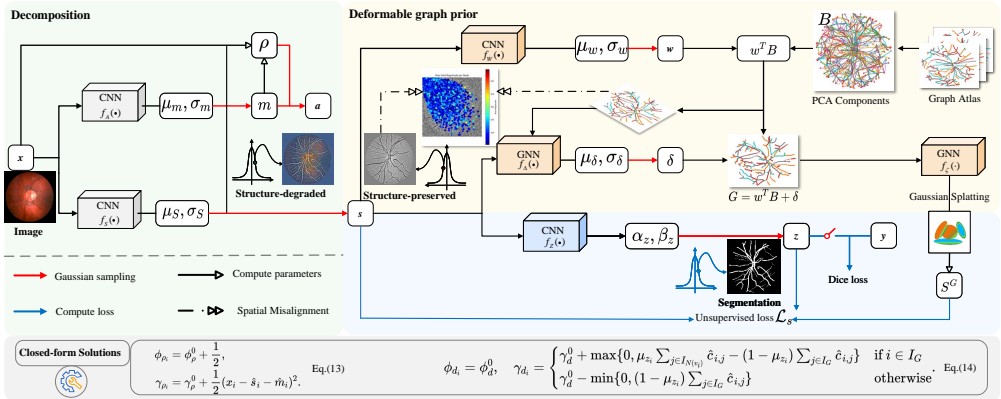

Figure 2: Network Architecture of GraphSeg. GraphSeg consists of three modules: image decomposition (green), vessel segmentation (blue) and deformable graph prior (yellow). During the training stage, the deformable graph prior guides both the decomposition and segmentation processes. In the inference stage, only decomposition and segmentation modules are used.

which not only has the same dimension as $S$, but also is differentiable. For convenience, the resulting Gaussian cloud is notated as $S^G = f_\zeta(G)$. Details are in Appendix A.2.5.

**Vascular Structure Matching.** Retinal vessels present geometric similarity between adjacent vascular segments but dissimilarity between landmarks and their neighbors, as shown in Fig. 1b. Motivated by this, we unfold $S \in \mathbb{R}^{h \times w}$ to $S_{unfold} \in \mathbb{R}^{hw \times k^2}$ using a sliding window of size $k \times k$, where each $k \times k$ matrix represents a local patch located at the central pixel. Similarly, we can unfold the Gaussian cloud $S^G$ as $S^G_{unfold}$. Let $\{v_i = (x_i, y_i)\}_{i=0}^{hw-1}$ denote the grid points corresponding to $S$, then these points can be partitioned into two groups based on whether they belong to the vertex set of deformable graph $G$. Concretely, $I_G = \{i | deg(v_i) > 0\}$ denote the indices of nodes with non-zero degree and $I_{N(v_i)}$ denotes the indices of $v_i$'s neighbors. Based on a all-to-all cosine similarity defined by $C = Sim_{\cos}(S_{unfold}, S^G_{unfold}) \in \mathbb{R}^{hw \times hw}$, the dissimilarity from $G$-to-neighbor can be defined as:

$$\mathcal{H}_{G2neighbor}(S; Z, G, D) = \sum_{i \in I_G} \sum_{j \in I_{N(v_i)}} z_i \cdot \exp\{-d_i \cdot c_{i,j}\}. \tag{1}$$

Here, $z_i \approx 1$ denotes the $i$-th pixel with a big probability belongs to vascular foreground. This loss forces to learn similar adjacent vascular segments while detect dissimilar landmarks. Similarly, the similarity from $G$-to-background can be defined as,

$$\mathcal{H}_{G2background}(S; Z, G, D) = \sum_{j \in I_G} \sum_{i=0}^{hw-1} (1 - z_i) \cdot \exp\{d_i \cdot c_{i,j}\}. \tag{2}$$

Here, $(1 - z_i) \approx 1$ denotes the $i$-th pixel with a big probability belongs to non-vascular background. This enforces our model to distinguish between vascular structure and non-vascular background. By combining both, we define an energy function by,

$$\mathcal{H}(S; Z, G, D) = \mathcal{H}_{G2neighbor}(S; Z, G, D) + \mathcal{H}_{G2background}(S; Z, G, D). \tag{3}$$

This aims to keep adjacent local shapes (if not landmarks) as similar as possible while distinguishing unconnected local shapes. Finally, we have,

$$p_\zeta(S | Z, G, D) \propto \exp\{-\mathcal{H}(S; Z, G, D)\}, \tag{4}$$

which informs a structure-preserved image decomposition and vessel segmentation, guided by the deformable graph prior.

### 3.3 Deep Variational Inference

**Maximum a *posteriori*.** Let $\Psi = \{\mathbf{s}, \mathbf{m}, \rho, \mathbf{z}, \mathbf{d}, \delta, \mathbf{w}\}$ denote the set of variables that need to be inferred, we aim to maximize $p_\zeta(\Psi | \mathbf{x}) \propto p(\mathbf{x} | \Psi) p_\zeta(\Psi)$, which however is intractable due to unknown

parameters $\zeta$. To tackle the difficulty, we adopt a variational distribution $q_\theta(\Psi|\mathbf{x})$ to approximate $p(\Psi|\mathbf{x})$, and maximize an evidence lower bound (ELBO) as follows,

$$\mathcal{L}(\zeta, \theta; \mathbf{x}) := \mathbb{E}_{\Psi \sim q_\theta(\Psi|\mathbf{x})} \left[ \ln \frac{p_\zeta(\Psi, \mathbf{x})}{q_\theta(\Psi|\mathbf{x})} \right]. \tag{5}$$

Since $p_\zeta(\Psi, \mathbf{x}) = p(\mathbf{x}|\Psi)p_\zeta(\Psi)$, the above equation can be converted to,

$$\mathcal{L}(\zeta, \theta; \mathbf{x}) := \mathbb{E}_{\Psi \sim q_\theta(\Psi|\mathbf{x})} [\ln p(\mathbf{x}|\Psi)] - \text{KL}(q_\theta(\Psi|\mathbf{x})\|p_\zeta(\Psi)). \tag{6}$$

The first term induces a reconstruction loss,

$$\mathbb{E}_{\Psi \sim q_\theta(\Psi|\mathbf{x})} [\ln p(\mathbf{x}|\Psi)] = \mathbb{E}_{q_\theta(\mathbf{s}, \mathbf{m}, \rho|\mathbf{x})} [\ln p(\mathbf{x}|\mathbf{s}, \mathbf{m}, \rho)]. \tag{7}$$

**KL Divergence**. Based on the PGM in Fig. 1a, the prior $p(\Psi)$ can be expressed as,

$$p_\zeta(\Psi) = p_\zeta(\mathbf{s}, \mathbf{m}, \rho, \mathbf{z}, \mathbf{d}, \delta, \mathbf{w}) = p_\zeta(\mathbf{s}|\mathbf{z}, \mathbf{d}, \delta, \mathbf{w})p(\mathbf{m})p(\rho)p(\mathbf{z})p(\mathbf{d})p(\delta)p(\mathbf{w}). \tag{8}$$

Next, we factorize the variational distribution $q_\theta(\Psi|\mathbf{x})$ as two distributions corresponding to the set of image-related variables, $\{\mathbf{s}, \mathbf{m}, \rho\}$, and the set of structure-related variables, $\{\mathbf{z}, \mathbf{d}, \delta, \mathbf{w}\}$,

$$q_\theta(\Psi|\mathbf{x}) = q_\theta(\mathbf{s}, \mathbf{m}, \rho, \mathbf{z}, \mathbf{d}, \delta, \mathbf{w}|\mathbf{x}) = q_\theta(\mathbf{s}, \mathbf{m}, \rho|\mathbf{x})q_\theta(\mathbf{z}, \mathbf{d}, \delta, \mathbf{w}|\mathbf{s}). \tag{9}$$

The distribution of the image-related variables can be further factorized as,

$$q_\theta(\mathbf{s}, \mathbf{m}, \rho|\mathbf{x}) = q_\theta(\mathbf{m}|\mathbf{x})q_\theta(\mathbf{s}|\mathbf{x}, \mathbf{m})q_\theta(\rho|\mathbf{x}, \mathbf{m}, \mathbf{s}). \tag{10}$$

Similarly, the distribution of the structure-related variables can be further factorized as,

$$q_\theta(\mathbf{z}, \mathbf{d}, \delta, \mathbf{w}|\mathbf{s}) = q_\theta(\mathbf{d}|\mathbf{s})q_\theta(\mathbf{w}|\mathbf{s}, \mathbf{d})q_\theta(\delta|\mathbf{s}, \mathbf{d}, \mathbf{w})q_\theta(\mathbf{z}|\mathbf{s}, \mathbf{d}, \delta, \mathbf{w}). \tag{11}$$

Based on the above factorization, the second KL divergence term in Eq. (6) can be unfolded as,

$$\begin{aligned}
\text{KL}(q_\theta(\Psi|\mathbf{x})\|p_\zeta(\Psi)) = {} & \mathbb{E}_{q_\theta(\mathbf{z}, \mathbf{d}, \delta, \mathbf{w}|\mathbf{s})} [\text{KL}(q_\theta(\mathbf{s}, \mathbf{m}, \rho|\mathbf{x})\|p_\zeta(\mathbf{s}|\mathbf{z}, \mathbf{d}, \delta, \mathbf{w})p(\mathbf{m})p(\rho))] \\
& + \mathbb{E}_{q_\theta(\mathbf{s}|\mathbf{x}, \mathbf{m})} [\text{KL}(q_\theta(\mathbf{z}, \mathbf{d}, \delta, \mathbf{w}|\mathbf{s})\|p(\mathbf{z})p(\mathbf{d})p(\delta)p(\mathbf{w}))].
\end{aligned} \tag{12}$$

**Closed-form Solutions.** Since $\rho$ follows a Gamma prior, its variational distribution also follows a Gamma distribution, i.e., $q_\theta(\rho|\mathbf{x}, \mathbf{m}, \mathbf{s}) = \mathcal{G}(\rho|\phi_\rho, \gamma_\rho)$. By minimizing the KL divergence over $q_\theta(\rho|\mathbf{x}, \mathbf{m}, \mathbf{s})$, we have the following closed-formed solution,

$$\phi_{\rho_i} = \phi_\rho^0 + \frac{1}{2}, \quad \gamma_{\rho_i} = \gamma_\rho^0 + \frac{1}{2}(x_i - \hat{s}_i - \hat{m}_i)^2. \tag{13}$$

Similarly, the variational distribution of $\mathbf{d}$ follows a Gamma distribution, i.e., $q_\theta(\mathbf{d}|\mathbf{s}) = \mathcal{G}(\mathbf{d}|\phi_d, \gamma_d)$. Its distributional parameters can be explicitly computed by,

$$\phi_{d_i} = \phi_d^0, \quad \gamma_{d_i} = \begin{cases} \gamma_d^0 + \max\{0, \mu_{z_i} \sum_{j \in I_{N(v_i)}} \hat{c}_{i,j} - (1 - \mu_{z_i}) \sum_{j \in I_G} \hat{c}_{i,j}\} & \text{if } i \in I_G \\ \gamma_d^0 - \min\{0, (1 - \mu_{z_i}) \sum_{j \in I_G} \hat{c}_{i,j}\} & \text{otherwise} \end{cases}. \tag{14}$$

Here, $\max / \min$ is used to ensure a feasible Gamma distribution, and $\hat{s}_i, \hat{m}_i, \hat{c}_{i,j}$ denotes obtained samples by Markov Chain Monte Carlo (MCMC) sampling.

**Learnable Variational Posteriors.** Given $q_\theta(\rho|\mathbf{x}, \mathbf{m}, \mathbf{s}) = \mathcal{G}(\rho|\phi_\rho, \gamma_\rho)$ and $q_\theta(\mathbf{d}|\mathbf{s}) = \mathcal{G}(\mathbf{d}|\phi_d, \gamma_d)$, the variational distributions of other variables can be learned by minimizing,

$$\begin{aligned}
\mathcal{L}(\zeta, \theta; \mathbf{x}, \rho, \mathbf{d}) &= \mathbb{E}_{q_\theta(\Psi|\mathbf{x})} [-\ln p(\mathbf{x}|\Psi)] + \text{KL}(q_\theta(\Psi|\mathbf{x})\|p_\zeta(\Psi)) \\
&= \underbrace{\mathcal{L}_x(\theta; \mathbf{x}, \rho) + \mathcal{L}_m(\theta)}_{\text{image-related}} + \underbrace{\mathcal{L}_s(\theta, \zeta; \mathbf{d}) + \mathcal{L}_z(\theta) + \mathcal{L}_\delta(\theta) + \mathcal{L}_\mathbf{w}(\theta)}_{\text{structure-related}}.
\end{aligned} \tag{15}$$

Please refer to Appendix A.2 for the detailed formulation of this section.

**Network Architecture.** We implement the variational inference framework with deep neural networks, as shown in Fig. 2. GraphSeg consists of three modules: image decomposition, deformable graph prior, and vessel segmentation. For the decomposition module, we employ two separate residual networks [30], denoted as $f_A$ and $f_S$, to infer the structure-degraded component $\mathbf{m}$ and the structure-preserved representation $\mathbf{s}$ of the input image $\mathbf{x}$. Both $\mathbf{m}$ and $\mathbf{s}$ are modeled as samples drawn from

Gaussian distributions, enabling a probabilistic interpretation of the decomposed features. For the deformable graph prior module, we first utilize a convolutional neural network $f_W$ to infer the latent principal component coefficients $\mathbf{w}$ from the input vascular structure representation $\mathbf{s}$. The misaligned graph template is then reconstructed as $\mathbf{w}^\top B$, and refined by a graph neural network (GNN) $f_\Delta$ to predict a displacement field $\delta \sim \mathcal{N}(\mu_\delta, \sigma_\delta^2)$. This results in a deformable graph topology $G$. For the segmentation module, a second graph neural network $f_\zeta$ is used to predict the local scale parameters of each node in $G$, producing the final graph-based vascular structure $S^G$ through Gaussian splatting. This result is correlated with the decomposed structure $S$ for downstream segmentation. A third U-Net decoder is used to infer the final segmentation map $\mathbf{z}$, which is modeled as a sample from a Beta distribution, $\mathcal{B}(\mathbf{z}|\alpha_z, \beta_z)$. After that, the unsupervised energy loss, $L_s(\theta, \zeta; \mathbf{d})$, is computed based on the consistency among the predicted segmentation $\mathbf{z}$, the graph-based structure $S^G$, the decomposed structure $S$, and the landmark $\mathbf{d}$, as formulated in Eq. (3), where $\mathbf{d}$ is obtained through a closed form as Eq. (14). Finally, we combine the Dice loss with the unsupervised loss in Eq. (15) by $\mathcal{L}(\zeta, \theta; \mathbf{x}, \rho, \mathbf{d}) + \mathcal{L}_{Dice}(\theta; \mathbf{z}, \mathbf{y})$ for training neural networks.

# 4 Experiments

To assess the performance and generalization capability of GraphSeg, we train it on a single dataset and evaluate it across all other datasets to measure its generalization effectiveness.

## 4.1 Setup

**Datasets and Metrics.** We employ four widely recognized datasets: CHASE [31], HRF [32], DRIVE [33], and STARE [34]. For evaluation, we mainly employ the Accuracy (**Acc**), **F1**, **Soft Dice**, **Sensitivity**, and **Specificity** to evaluate the segmentation performance. **Acc**, **F1**, and **Soft Dice** are used to measure the overall segmentation performance, while **Sensitivity** and **Specificity** are used to measure the performance of vascular foreground and non-vascular background, respectively.

**Compared Approaches.** We mainly compare GraphSeg with the following representative methods: (1) previous state-of-the-art vessel segmentation models, including FR-Net [5] and FSG-Net [11]; (2) widely adopted architectures for medical image segmentation, such as U-Net [35], AttUNet [36], AGNet [37], ConvUNeXt [38], DCSAU-Net [39], R2UNet [40], and SAUNet [41]; (3) a strong and generalizable baseline, BayeSeg [8]; and (4) Skelcon [4], which attempts to introduce the skeleton structural prior into retinal vessel segmentation. *To ensure a fair comparison in generalization ability*, both U-Net and BayeSeg are re-implemented with the same segmentation backbone used in GraphSeg to eliminate architecture-induced bias. The graph construction follows Bal et al. [42]. Graph Atlas is obtained by performing their code on the training set of CHASE. More implementation details can be found in the Appendix B.

## 4.2 Experimental Results

We first compare GraphSeg with state-of-the-art methods in terms of accuracy, sensitivity, specificity, F1 score, and Soft Dice. To evaluate cross-domain generalization, the models are trained on one dataset and tested on the others. Finally, we conduct an ablation study to quantify the contribution of each component within GraphSeg.

### 4.2.1 Comparison with Previous Models

To evaluate the effectiveness of GraphSeg on retinal vessel segmentation, we first compare our Graph-Seg with previous models by training and testing on the same dataset. As shown in Table 1, under the same segmentation backbone, both BayeSeg and GraphSeg outperform U-Net across all datasets, highlighting the effectiveness of the variational decomposition in improving vascular segmentation. Furthermore, GraphSeg consistently surpasses BayeSeg, especially on structure-sensitive metrics such as Soft Dice and F1 score, demonstrating the additional benefit of incorporating a graph-based structural prior. Compared with other state-of-the-art methods (e.g., FR-Net, FSG-Net, Skelton), GraphSeg achieves competitive or superior results, confirming that our framework is comparable to specialized architectures while offering stronger anatomical consistency and generalization potential. Without loss of generality, any other models, including the FR-Net and FSG-Net, can be integrated into our GraphSeg framework for graph prior injection.

Table 1: Performance comparison of different methods across CHASE, DRIVE, and HRF datasets. The best results per dataset are in **bold**. *Note that U-Net, BayeSeg, and GraphSeg used the same segmentation backbone.*

| Dataset | Method | Accuracy | Soft Dice | F1 | Sensitivity | Specificity |
|---------|--------|----------|-----------|-----|-------------|-------------|
| CHASE | Skelton | 95.61 | 80.71 | 78.95 | 78.17 | 97.94 |
| | FR-Net | 97.26 | 80.10 | 79.10 | 84.74 | 97.65 |
| | FSG-Net | **97.52** | 81.34 | 81.02 | 86.00 | **98.26** |
| | U-Net | 95.17 | 79.74 | 78.39 | 81.97 | 96.79 |
| | BayeSeg | 95.45 | 80.48 | 79.62 | 82.93 | 96.99 |
| | **GraphSeg** | 96.54 | **82.91** | **81.82** | **87.84** | 97.40 |
| DRIVE | Skelton | 94.61 | 81.50 | 80.39 | 83.23 | 98.59 |
| | FR-Net | 97.01 | 83.91 | 82.99 | 83.86 | 98.15 |
| | FSG-Net | **97.04** | 84.19 | 83.23 | **84.21** | 98.30 |
| | U-Net | 90.89 | 61.09 | 61.09 | 48.55 | **98.51** |
| | BayeSeg | 94.48 | 79.08 | 78.56 | 77.60 | 97.06 |
| | **GraphSeg** | 96.13 | **85.23** | **84.82** | 83.02 | 98.15 |
| HRF | Skelton | 95.90 | 79.87 | 78.59 | 78.53 | 98.86 |
| | FR-Net | 97.01 | 81.92 | 80.70 | 81.67 | 98.42 |
| | FSG-Net | **97.10** | 82.51 | 81.57 | **83.61** | **98.99** |
| | U-Net | 89.79 | 71.37 | 67.76 | 78.15 | 91.74 |
| | BayeSeg | 95.35 | 81.77 | 81.05 | 75.44 | 98.42 |
| | **GraphSeg** | 95.88 | **84.29** | **83.89** | 81.58 | 98.07 |

### 4.2.2 Cross-domain Generalization

**Cross-dataset Generalizability.** To assess the generalizability of GraphSeg, we perform cross-dataset evaluations by training the model on the CHASE dataset and testing it on three unseen datasets: DRIVE, HRF and STARE. As shown in Table 2, we compare against ten strong baselines. *To ensure a fair comparison*, U-Net, BayeSeg, and GraphSeg are trained with *the same data augmentation* and implemented with *the same segmentation backbone*, eliminating potential confounding factors from training protocols or model capacity. In particular, U-Net is trained and tested on each dataset independently, serving as an in-domain reference. BayeSeg and GraphSeg are trained on CHASE and evaluated on all test sets to assess their cross-domain performance.

Across domains, BayeSeg and GraphSeg consistently outperform others in Soft Dice, demonstrating the advantage of image decomposition in improving robustness under distribution shift. Notably, GraphSeg outperforms the second best by 7.43 in Soft Dice and 6.46 in F1 Score on the HRF dataset, where the background is more complex and noisy, highlighting the importance of incorporating a graph structure prior to cross-domain generalization. Compared with the in-domain results in Table 1, although BayeSeg and GraphSeg present significant performance drops due the low-resolution nature of DRIVE, which raises difficulties in decomposing vascular structures, GraphSeg still performs much better on DRIVE, demonstrating the effectiveness of statistical modeling of vsacular structures.

**Generalizability on Cross-dataset Graph Prior.** We also use the graph prior from the CHASE to train GraphSeg on the DRIVE dataset, as shown in Table 3. The results are summarized as follows: (1) GraphSeg (trained) outperforms U-Net (trained) due to the graph prior, which demonstrates that the graph prior extracted from one dataset can be transferred to the other dataset. (2) GraphSeg outperforms U-Net (trained) on most metrics, which confirms the generalizability of GraphSeg and demonstrates the robustness of our framework with respect to graph priors.

### 4.3 Ablation Study

To evaluate the impact of the graph prior and the image decomposition module, we conduct an ablation study by training the models on the CHASE dataset and testing them on the DRIVE dataset. Both the two modules contribute positively to the generalization performance as shown in Table 4.

Table 2: Cross-dataset evaluation: Models are trained on CHASE and evaluated on other datasets. U-Net is trained and tested on each dataset as a strong baseline.

| Dataset | Model | Acc | Soft Dice | F1 | Sensitivity | Specificity |
|---|---|---|---|---|---|---|
| **CHASE (train)** | U-Net | 95.17 | 79.74 | 78.39 | 81.97 | 96.79 |
| | FSGNet | 97.52 | 81.34 | 81.02 | 86.00 | 98.26 |
| | FRNet | 97.26 | 80.10 | 79.10 | 84.74 | 97.65 |
| | AttUNet | **97.54** | 66.02 | 77.06 | 78.92 | **98.56** |
| | AGNet | 97.44 | 76.25 | 77.58 | 85.14 | 98.13 |
| | ConvUNeXt | 97.21 | 72.40 | 73.93 | 75.97 | 98.39 |
| | DCSAU-net | 97.17 | 74.33 | 74.68 | 79.87 | 98.12 |
| | R2UNet | 96.96 | 67.99 | 71.73 | 74.60 | 98.20 |
| | SAUNet | 97.47 | 76.17 | 77.23 | 82.02 | 98.33 |
| | BayeSeg | 95.45 | 80.48 | 79.62 | 82.93 | 96.99 |
| | GraphSeg | 96.54 | **82.91** | **81.82** | **87.84** | 97.40 |
| **DRIVE** | U-Net (trained) | 90.89 | 61.09 | 61.09 | 48.55 | 98.51 |
| | FSGNet | 95.62 | 57.51 | 57.56 | 44.39 | **99.46** |
| | FRNet | 96.65 | 71.33 | 73.57 | 67.53 | 98.85 |
| | AttUNet | 95.07 | 45.27 | 50.22 | 38.75 | 99.30 |
| | AGNet | 96.25 | 69.46 | 70.59 | 55.21 | 98.59 |
| | ConvUNeXt | **97.07** | 61.91 | 63.77 | **74.35** | 97.92 |
| | DCSAU-net | 95.30 | 62.30 | 62.56 | 56.82 | 98.19 |
| | R2UNet | 95.01 | 63.86 | 67.02 | 70.14 | 96.60 |
| | SAUNet | 96.46 | 69.78 | 70.67 | 61.92 | 99.06 |
| | BayeSeg | 92.34 | 71.68 | 70.71 | 61.85 | 97.87 |
| | GraphSeg | 93.56 | **77.16** | **76.01** | 68.09 | 98.19 |
| **HRF** | U-Net (trained) | 89.79 | 71.37 | 67.76 | 78.15 | 91.74 |
| | FSGNet | 95.90 | 68.26 | 68.56 | 71.50 | 98.28 |
| | FRNet | 97.00 | 64.99 | 67.56 | 69.78 | 97.28 |
| | AttUNet | **97.37** | 54.20 | 65.29 | 66.41 | 97.94 |
| | AGNet | 96.70 | 63.03 | 64.54 | 66.69 | 97.10 |
| | ConvUNeXt | 96.52 | 59.74 | 60.04 | 63.58 | **99.07** |
| | DCSAU-net | 96.63 | 61.70 | 62.09 | 68.56 | 97.30 |
| | R2UNet | 94.21 | 49.07 | 52.03 | **88.87** | 94.40 |
| | SAUNet | 97.15 | 65.87 | 67.17 | 83.92 | 97.65 |
| | BayeSeg | 92.05 | 70.73 | 70.13 | 67.71 | 96.13 |
| | GraphSeg | 93.38 | **78.16** | **76.59** | 78.90 | 96.88 |
| **STARE** | U-Net (trained) | 90.98 | 60.81 | 59.66 | 55.71 | 95.95 |
| | FSGNet | 96.15 | 54.08 | 54.23 | 45.43 | 99.33 |
| | FRNet | 93.04 | 64.91 | 67.07 | 62.39 | 98.96 |
| | AttUNet | 95.71 | 37.91 | 41.97 | 34.39 | **99.54** |
| | AGNet | 96.52 | 67.49 | 68.82 | 66.74 | 98.47 |
| | ConvUNeXt | 95.97 | 52.65 | 53.30 | 46.89 | 98.96 |
| | DCSAU-net | 94.77 | 47.59 | 47.83 | 45.65 | 97.75 |
| | R2UNet | 95.53 | 49.90 | 50.86 | 50.63 | 98.22 |
| | SAUNet | **96.87** | 61.50 | 62.23 | 56.44 | 99.29 |
| | BayeSeg | 92.85 | 67.82 | 66.82 | 59.78 | 97.65 |
| | GraphSeg | 93.65 | **72.76** | **71.43** | **68.72** | 97.69 |

Introducing the decomposition module alone brings consistent improvements across all metrics, particularly in sensitivity and Dice score, indicating enhanced structural representation. When the graph prior is additionally incorporated, the model achieves the best performance, demonstrating the complementary role of topological constraints in further improving generalization to unseen data.

Table 3: Evaluation of model generalizability on the cross-dataset graph prior

| Dataset | Model | Acc | Soft Dice | F1 | Sensitivity | Specificity |
|---|---|---|---|---|---|---|
| **DRIVE (train)** | GraphSeg (trained) | 96.13 | 85.23 | 84.82 | 83.02 | 98.15 |
| | U-Net (trained) | 90.89 | 61.09 | 61.09 | 48.55 | 98.51 |
| **CHASE** | GraphSeg | 95.02 | 82.56 | 79.37 | 89.70 | 95.69 |
| | U-Net (trained) | 95.17 | 79.74 | 78.39 | 81.97 | 96.79 |
| | GraphSeg (trained) | 96.54 | 82.91 | 81.82 | 87.84 | 97.40 |
| **HRF** | GraphSeg | 93.91 | 78.20 | 77.25 | 78.40 | 96.34 |
| | U-Net(trained) | 89.79 | 71.37 | 67.76 | 78.15 | 91.74 |
| | GraphSeg(trained) | 95.88 | 84.29 | 83.89 | 81.58 | 98.07 |

Table 4: Ablation on the effect of graph prior and image decomposition to generalizability.

| Decomposition | Graph prior | **Accuracy** | **Soft Dice** | **F1** | **Sensitivity** | **Specificity** |
|---|---|---|---|---|---|---|
| | | 90.75 | 66.56 | 64.07 | 55.04 | 97.19 |
| ✓ | | 92.34 | 71.68 | 70.71 | 61.85 | 97.87 |
| ✓ | ✓ | **93.56** | **77.16** | **76.01** | **68.09** | **98.19** |

## 5 Discussion

**Why does GraphSeg work?** The success of GraphSeg can be attributed to its unique integration of graph structure prior and image decomposition, which is modeled with an energy function as Eq. (3). The graph structure prior effectively captures the anatomical coherence of vascular trees, allowing the model to leverage both local connectivity and global shape constraints. This enables GraphSeg to achieve high-quality segmentation even in challenging scenarios, like complex noisy backgrounds. The image decomposition module further enhances this by disentangling the structure-preserved component from artifacts, leading to improved segmentation accuracy. Moreover, the introduction of an unsupervised energy function in Eq. (3) enables a natural extension of GraphSeg to semi-supervised learning settings, which are both practical and significant in real-world medical applications. We leave this direction as promising future work.

**How does GraphSeg generalize?** The generalizability of GraphSeg stems from two key factors: (1) the use of a graph structure prior that captures the underlying anatomy of retinal vessels, allowing the model to learn robust representations that are less sensitive to domain shifts; (2) the incorporation of a variational Bayesian framework that enables the model to adaptively learn stochastic intermediate samples from variational distributions, enhancing its ability to generalize to unseen domains. This combination allows GraphSeg to maintain high performance across different datasets.

## 6 Conclusion

In this work, we proposed GraphSeg, a variational Bayesian segmentation framework that integrates a deformable graph prior to anatomically consistent retinal vessel segmentation. By jointly modeling image decomposition and graph-aligned structural inference, GraphSeg captures both local connectivity and global vascular topology. Extensive experiments across CHASE, DRIVE, HRF, and STARE datasets demonstrate state-of-the-art accuracy and strong cross-domain generalization. In particular, our method consistently outperforms prior work in structure-sensitive metrics such as Soft Dice and F1, especially in challenging domains like HRF. Ablation studies further confirm the complementary roles of the graph prior and image decomposition. Overall, GraphSeg provides a principled and generalizable approach for structure-aware medical image segmentation. **Limitation:** While GraphSeg demonstrates strong performance and cross-domain generalization, its current formulation requires a pre-processing step to extract skeletons and construct graphs from segmentation masks. This dependency may introduce slight variations due to heuristic rules (e.g., junction merging, branch pruning), and future work could explore end-to-end learnable graph extraction to reduce reliance on hand-crafted morphological pipelines.

## Acknowledgments and Disclosure of Funding

This work was supported by the National Natural Science Foundation of China (61971142, 62111530195 and 62011540404), the development fund for Shanghai talents (2020015), and the Shanghai Municipal Education Commission-Artificial Intelligence Initiative to Promote Research Paradigm Reform and Empower Disciplinary Advancement Plan (grant no. 24KXZNA13).

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

# A Method Details

## A.1 Graph Construction and Shape PCA

The graph construction and shape PCA are conducted on the training set of CHASE dataset in this work. Given the segmentation masks, we construct a graph-based representation as our graph structural prior. Following Bal et al. [42] and Luo et al. [43], The process involves the following steps: (1) Skeletonization: We apply morphological thinning to obtain a 1-pixel-wide vessel skeleton, which preserves the topological structure while discarding width information. (2) Graph Construction: Nodes are extracted by identifying endpoints and bifurcations on the skeleton based on pixel connectivity. Edges are defined as pixel chains connecting node pairs, resulting in an undirected graph represented by an adjacency matrix. (3) Graph Refinement: To address artifacts such as node fragmentation and false intersections, we apply topological corrections by merging closely located nodes and pruning spurious connections based on angular continuity. (4) Graph Registration: Each graph is aligned to a common mean template using factorized graph matching [44]. Both node similarity (Euclidean distance) and edge similarity (shape distance computed via Square Root Velocity Functions, SRVFs [45]) are considered. (5) Shooting Vector Computation: The deformation from the mean graph $G_\mu$ to each registered graph $G_k^*$ is encoded as a shooting vector $sv_k = G_k^* - G_\mu$. (6) Feature Vector Construction: We concatenate the shooting vector and node coordinate differences into a high-dimensional vector $sv_k^*$ to capture both topological and geometric variability. (7) Principal Component Analysis. PCA is applied to the set $\{sv_k^*\}_{k=1}^m$ to obtain a compact shape representation. The resulting PCA coefficients characterize structural variation and are used in GraphSeg.

## A.2 Deep Variational Inference Details

### A.2.1 Derivation of Variational Inference

Let $\Psi = \{\mathbf{s}, \mathbf{m}, \rho, \mathbf{z}, \mathbf{d}, \delta, \mathbf{w}\}$ denote the set of variables that need to be inferred, we aim to maximize $p_\zeta(\Psi|\mathbf{x}) \propto p(\mathbf{x}|\Psi)p_\zeta(\Psi)$. We adopt a variational distribution $q_\theta(\Psi|\mathbf{x})$ to approximate $p(\Psi|\mathbf{x})$. As a result, this can be optimized by maximizing an evidence lower bound (ELBO) as follows,

$$\mathcal{L}(\zeta, \theta; \mathbf{x}) := \mathbb{E}_{\Psi \sim q_\theta(\Psi|\mathbf{x})}\left[\ln \frac{p_\zeta(\Psi, \mathbf{x})}{q_\theta(\Psi|\mathbf{x})}\right]. \tag{16}$$

Since $p_\zeta(\Psi, \mathbf{x}) = p(\mathbf{x}|\Psi)p_\zeta(\Psi)$, the above equation can be converted to

$$\mathcal{L}(\zeta, \theta; \mathbf{x}) := \mathbb{E}_{\Psi \sim q_\theta(\Psi|\mathbf{x})}[\ln p(\mathbf{x}|\Psi)] - \mathrm{KL}(q_\theta(\Psi|\mathbf{x})\|p_\zeta(\Psi)), \tag{17}$$

where the first term induces a reconstruction loss:

$$\mathbb{E}_{\Psi \sim q_\theta(\Psi|\mathbf{x})}[\ln p(\mathbf{x}|\Psi)] = \mathbb{E}_{q_\theta(\mathbf{s}, \mathbf{m}, \rho|\mathbf{x})}[\ln p(\mathbf{x}|\mathbf{s}, \mathbf{m}, \rho)]. \tag{18}$$

Based on the PGM as shown in Fig. 1a, the prior $p(\Psi)$ can be expressed as:

$$p_\zeta(\Psi) = p_\zeta(\mathbf{s}, \mathbf{m}, \rho, \mathbf{z}, \mathbf{d}, \delta, \mathbf{w}) = p_\zeta(\mathbf{s}|\mathbf{z}, \mathbf{d}, \delta, \mathbf{w})p(\mathbf{m})p(\rho)p(\mathbf{z})p(\mathbf{d})p(\delta)p(\mathbf{w}). \tag{19}$$

**Variational Distributions**. The variational distribution $q_\theta(\Psi|\mathbf{x})$ can be factorized as two distributions corresponding to the set of image-related variables, $\{\mathbf{s}, \mathbf{m}, \rho\}$, and the set of structure-related variables, $\{\mathbf{z}, \mathbf{d}, \delta, \mathbf{w}\}$,

$$q_\theta(\Psi|\mathbf{x}) = q_\theta(\mathbf{s}, \mathbf{m}, \rho, \mathbf{z}, \mathbf{d}, \delta, \mathbf{w}|\mathbf{x}) = q_\theta(\mathbf{s}, \mathbf{m}, \rho|\mathbf{x})q_\theta(\mathbf{z}, \mathbf{d}, \delta, \mathbf{w}|\mathbf{s}). \tag{20}$$

The distribution of the image-related variables can be further factorized by,

$$q_\theta(\mathbf{s}, \mathbf{m}, \rho|\mathbf{x}) = q_\theta(\mathbf{m}|\mathbf{x})q_\theta(\mathbf{s}|\mathbf{x}, \mathbf{m})q_\theta(\rho|\mathbf{x}, \mathbf{m}, \mathbf{s}). \tag{21}$$

That is we first estimate the mean $\mathbf{m}$ of $\mathbf{a}$ and then infer the vascular shape, $\mathbf{s}$, from $\mathbf{x}, \mathbf{m}$, and finally infer the inverse variance $\rho$ from $\mathbf{x}, \mathbf{m}, \mathbf{s}$. Similarly, the distribution of the structure-related variables can be further factorized by,

$$q_\theta(\mathbf{z}, \mathbf{d}, \delta, \mathbf{w}|\mathbf{s}) = q_\theta(\mathbf{d}|\mathbf{s})q_\theta(\mathbf{z}, \delta, \mathbf{w}|\mathbf{s}, \mathbf{d}) \tag{22}$$

$$= q_\theta(\mathbf{d}|\mathbf{s})q_\theta(\delta, \mathbf{w}|\mathbf{s}, \mathbf{d})q_\theta(\mathbf{z}|\mathbf{s}, \mathbf{d}, \delta, \mathbf{w}) \tag{23}$$

$$= q_\theta(\mathbf{d}|\mathbf{s})q_\theta(\mathbf{w}|\mathbf{s}, \mathbf{d})q_\theta(\delta|\mathbf{s}, \mathbf{d}, \mathbf{w})q_\theta(\mathbf{z}|\mathbf{s}, \mathbf{d}, \delta, \mathbf{w}). \tag{24}$$

That is we detect landmarks, $\mathbf{d}$, from the vascular shape $\mathbf{s}$, estimate the linear weights $\mathbf{w}$ which determines a vascular template $\mathbf{w}^T B$, infer displacement field $\delta$ between the vascular shape and template, and predict the segmentation $\mathbf{z}$ from the shape, landmark, and estimated graph $\mathbf{w}^T B + \delta$.

Based on the above factorizations, the second KL divergence term in (17) can be unfolded as,

$$\mathrm{KL}(q_\theta(\Psi|\mathbf{x})\|p_\zeta(\Psi)) = \mathbb{E}_{q_\theta(\mathbf{z},\mathbf{d},\delta,\mathbf{w}|\mathbf{s})} \left[\mathrm{KL}(q_\theta(\mathbf{s},\mathbf{m},\rho|\mathbf{x})\|p_\zeta(\mathbf{s}|\mathbf{z},\mathbf{d},\delta,\mathbf{w})p(\mathbf{m})p(\rho))\right] \tag{25}$$

$$+ \mathbb{E}_{q_\theta(\mathbf{s}|\mathbf{x},\mathbf{m})} \left[\mathrm{KL}(q_\theta(\mathbf{z},\mathbf{d},\delta,\mathbf{w}|\mathbf{s})\|p(\mathbf{z})p(\mathbf{d})p(\delta)p(\mathbf{w}))\right]. \tag{26}$$

### A.2.2 Closed-form Solution of $\rho$

Given other variables, $q_\theta(\rho|\mathbf{x},\mathbf{m},\mathbf{s}) = \mathcal{G}(\rho|\phi_\rho,\gamma_\rho)$ can be inferred by minimizing,

$$\mathcal{L}(\phi_\rho,\gamma_\rho) = \mathbb{E}\left[-\ln p(\mathbf{x}|\mathbf{s},\mathbf{m},\rho)\right] + \mathbb{E}\left[\mathrm{KL}(q_\theta(\rho|\mathbf{x},\mathbf{m},\mathbf{s})\|p(\rho))\right] + C \tag{27}$$

$$= \mathbb{E}_{q_\theta(\rho|\mathbf{x},\mathbf{m},\mathbf{s})}\left[\ln q_\theta(\rho|\mathbf{x},\mathbf{m},\mathbf{s})\right] - \mathbb{E}_{q_\theta(\rho|\mathbf{x},\mathbf{m},\mathbf{s})}\left[\ln p(\rho)\right. \tag{28}$$

$$\left. + \mathbb{E}_{q_\theta(\Psi\backslash\rho|\mathbf{x})}\left[\ln p(\mathbf{x}|\mathbf{s},\mathbf{m},\rho)\right] + C\right]. \tag{29}$$

The minimum is achieved when

$$\ln q_\theta(\rho|\mathbf{x},\mathbf{m},\mathbf{s}) = \ln p(\rho) + \mathbb{E}_{q_\theta(\Psi\backslash\rho|\mathbf{x})}\left[\ln p(\mathbf{x}|\mathbf{s},\mathbf{m},\rho)\right] + C \tag{30}$$

$$\approx \sum_{i=0}^{hw-1}\left[(\phi_\rho^0 - 1)\ln\rho_i - \gamma_\rho^0\rho_i\right] \tag{31}$$

$$+ \sum_{i=0}^{hw-1}\frac{1}{2}\left[\ln\rho_i - (x_i - \hat{s}_i - \hat{m}_i)^2\rho_i\right] + C \tag{32}$$

$$= \sum_{i=0}^{hw-1}\left[(\phi_\rho^0 - \frac{1}{2})\ln\rho_i - (\gamma_\rho^0 + \frac{1}{2}(x_i - \hat{s}_i - \hat{m}_i)^2)\rho_i\right] + C. \tag{33}$$

Here, we use Markov Chain Monte Carlo (MCMC) sampling in (32) to approximate the expectation in (30). Since $q_\theta(\rho|\mathbf{x},\mathbf{m},\mathbf{s}) = \mathcal{G}(\rho|\phi_\rho,\gamma_\rho)$, we finally have

$$\phi_{\rho_i} = \phi_\rho^0 + \frac{1}{2}, \quad \gamma_{\rho_i} = \gamma_\rho^0 + \frac{1}{2}(x_i - \hat{s}_i - \hat{m}_i)^2. \tag{34}$$

### A.2.3 Closed-form Solution of $d$

Given other variables, $q_\theta(\mathbf{d}|\mathbf{s}) = \mathcal{G}(\mathbf{d}|\phi_d,\gamma_d)$ can be inferred by minimizing,

$$\mathcal{L}(\phi_d,\gamma_d) = \mathbb{E}\left[-\ln p_\zeta(\mathbf{s}|\mathbf{z},\mathbf{d},\delta,\mathbf{w})\right] + \mathbb{E}\left[\mathrm{KL}(q_\theta(\mathbf{d}|\mathbf{s})\|p(\mathbf{d}))\right] + C \tag{35}$$

$$= \mathbb{E}_{q_\theta(\mathbf{d}|\mathbf{s})}\left[\ln q_\theta(\mathbf{d}|\mathbf{s})\right] - \mathbb{E}_{q_\theta(\mathbf{d}|\mathbf{s})}\left[\ln p(\mathbf{d}) + \mathbb{E}_{q_\theta(\Psi\backslash\mathbf{d}|\mathbf{x})}\left[\ln p_\zeta(\mathbf{s}|\mathbf{z},\mathbf{d},\delta,\mathbf{w})\right] + C\right]. \tag{36}$$

The minimum is achieved when

$$\ln q_\theta(\mathbf{d}|\mathbf{s}) = \ln p(\mathbf{d}) + \mathbb{E}_{q_\theta(\Psi\backslash\mathbf{d}|\mathbf{x})}\left[\ln p_\zeta(\mathbf{s}|\mathbf{z},\mathbf{d},\delta,\mathbf{w})\right] + C \tag{37}$$

$$\approx \ln p(\mathbf{d}) + \sum_{i\in I_G}\sum_{j\in I_{N(v_i)}}\mu_{z_i}\cdot\exp\{-d_i\cdot\hat{c}_{i,j}\} + \sum_{j\in I_G}\sum_{i=0}^{hw-1}(1-\mu_{z_i})\cdot\exp\{d_i\cdot\hat{c}_{i,j}\} + C \tag{38}$$

$$= \sum_{i=0}^{hw-1}\left[(\phi_d^0 - 1)\ln d_i - \gamma_d^0 d_i\right] \tag{39}$$

$$+ \sum_{i\in I_G}\sum_{j\in I_{N(v_i)}}\mu_{z_i}\cdot\exp\{-d_i\cdot\hat{c}_{i,j}\} + \sum_{j\in I_G}\sum_{i=0}^{hw-1}(1-\mu_{z_i})\cdot\exp\{d_i\cdot\hat{c}_{i,j}\} + C, \tag{40}$$

where $\mu_{z_i} = \alpha_{z_i}/(\alpha_{z_i} + \beta_{z_i})$, and we approximate the expectation at the right-hand of (37) by the MCMC sampling in (38). Furthermore, since $e^{\lambda x} = 1 + \lambda x + O(x^2)$, we have $\exp\{-d_i\cdot\hat{c}_{i,j}\} \approx$

$1 - d_i \cdot \hat{c}_{i,j}$ and $\exp\{d_i \cdot \hat{c}_{i,j}\} \approx 1 + d_i \cdot \hat{c}_{i,j}$. As a result, we can convert the above formula to,

$$\ln q_\theta(\mathbf{d}|\mathbf{s}) \approx \sum_{i \in I_G} \left[ (\phi_d^0 - 1) \ln d_i - \left( \gamma_d^0 + \mu_{z_i} \sum_{j \in I_{N(v_i)}} \hat{c}_{i,j} - (1 - \mu_{z_i}) \sum_{j \in I_G} \hat{c}_{i,j} \right) d_i \right] \quad (41)$$

$$+ \sum_{i \notin I_G} \left[ (\phi_d^0 - 1) \ln d_i - \left( \gamma_d^0 - (1 - \mu_{z_i}) \sum_{j \in I_G} \hat{c}_{i,j} \right) d_i \right] + C. \quad (42)$$

Thus, the parameters of $q_\theta(\mathbf{d}|\mathbf{s}) = \mathcal{G}(\mathbf{d}|\phi_d, \gamma_d)$ can be explicitly expressed as,

$$\phi_{d_i} = \phi_d^0, \quad \gamma_{d_i} = \begin{cases} \gamma_d^0 + \max\{0, \mu_{z_i} \sum_{j \in I_{N(v_i)}} \hat{c}_{i,j} - (1 - \mu_{z_i}) \sum_{j \in I_G} \hat{c}_{i,j}\} & \text{if } i \in I_G \\ \gamma_d^0 - \min\{0, (1 - \mu_{z_i}) \sum_{j \in I_G} \hat{c}_{i,j}\} & \text{otherwise} \end{cases}. \quad (43)$$

Here, $\max / \min$ is used to ensure a feasible Gamma distribution.

### A.2.4  Unsupervised Loss Details

In Eq. (15), the unsupervised loss consists of image-related parts and structure-related parts. To be specific, we describe the terms as follows:

$$\mathcal{L}_x(\theta; \mathbf{x}, \rho) = \mathbb{E}_{q_\theta(\mathbf{s}, \mathbf{m}, \rho|\mathbf{x})} \left[ -\ln p(\mathbf{x}|\mathbf{s}, \mathbf{m}, \rho) \right] \quad (44)$$

$$= \mathbb{E}_{q_\theta(\mathbf{m}|\mathbf{x}) q_\theta(\mathbf{s}|\mathbf{x}, \mathbf{m}) q_\theta(\rho|\mathbf{x}, \mathbf{m}, \mathbf{s})} \left[ -\ln p(\mathbf{x}|\mathbf{s}, \mathbf{m}, \rho) \right] \quad (45)$$

$$= \mathbb{E}_{q_\theta(\mathbf{m}|\mathbf{x}) q_\theta(\mathbf{s}|\mathbf{x}, \mathbf{m})} \left[ \frac{1}{2} \sum_{i=0}^{hw-1} \mu_{\rho_i}(x_i - s_i - m_i)^2 \right] + C \quad (46)$$

$$\approx \frac{1}{2} \|x - \hat{s} - \hat{m}\|_{diag(\mu_\rho)}^2 + C, \quad (47)$$

where $\mu_{\rho_i} = \frac{\phi_{\rho_i}}{\gamma_{\rho_i}}$ denotes the mean of $\rho_i$. Suppose $q_\theta(\mathbf{m}) = \mathcal{N}(\mathbf{m}|\mu_m, diag(\sigma_m^2))$, then

$$\mathcal{L}_m(\theta) = \mathbb{E}_{q_\theta(\mathbf{m}|\mathbf{x})} \left[ \text{KL}(q_\theta(\mathbf{m}|\mathbf{x})||p(\mathbf{m})) \right] \quad (48)$$

$$= \mathbb{E}_{q_\theta(\mathbf{m}|\mathbf{x})} \left[ \ln q_\theta(\mathbf{m}|\mathbf{x}) \right] + \mathbb{E}_{q_\theta(\mathbf{m}|\mathbf{x})} \left[ -\ln p(\mathbf{m}) \right] \quad (49)$$

$$= \frac{1}{2} \sigma_m^0 \|\mu_m - \mu_m^0\|_2^2 + \frac{1}{2}[\langle \sigma_m^0 \mathbf{1}, \sigma_m^2 \rangle - \langle \mathbf{1}, \ln \sigma_m^2 \rangle] + C. \quad (50)$$

Suppose $q_\theta(\mathbf{s}|\mathbf{x}, \mathbf{m}) = \mathcal{N}(\mathbf{s}|\mu_s, diag(\sigma_s^2))$, then

$$\mathcal{L}_s(\theta, \zeta; \mathbf{d}) = \mathbb{E}_{q_\theta(\mathbf{z}, \mathbf{d}, \delta, \mathbf{w}|\mathbf{s})} \left[ \text{KL}(q_\theta(\mathbf{s}|\mathbf{x}, \mathbf{m})||p_\zeta(\mathbf{s}|\mathbf{z}, \mathbf{d}, \delta, \mathbf{w})) \right] \quad (51)$$

$$= \mathbb{E}_{q_\theta(\mathbf{s}|\mathbf{x}, \mathbf{m})} \left[ \ln q_\theta(\mathbf{s}|\mathbf{x}, \mathbf{m}) \right] + \mathbb{E}_{q_\theta(\mathbf{s}|\mathbf{x}, \mathbf{m}) q_\theta(\mathbf{z}, \mathbf{d}, \delta, \mathbf{w}|\mathbf{s})} \left[ -\ln p_\zeta(\mathbf{s}|\mathbf{z}, \mathbf{d}, \delta, \mathbf{w}) \right] \quad (52)$$

$$= \sum_{i \in I_G} \sum_{j \in I_{N(v_i)}} \mu_{z_i} \cdot M_{d_i}(-\hat{c}_{i,j}) + \sum_{j \in I_G} \sum_{i=0}^{hw-1} (1 - \mu_{z_i}) \cdot M_{d_i}(\hat{c}_{i,j}) - \sum_{i=0}^{hw-1} \ln \sigma_s^2 + C, \quad (53)$$

where, $M_{d_i}(t) = (1 - \frac{t}{\gamma_{d_i}})^{-\phi_{d_1}}$ (for $t < \gamma_{d_i}$) denotes the moment-generating function of $d_i$. In practical implementation, we need to clip $\hat{c}_{i,j}$ and ensure it satisfy $-\gamma_{d_i} + \epsilon \leq \hat{c}_{i,j} \leq \gamma_{d_i} - \epsilon$. Suppose $q_\theta(\mathbf{z}|\mathbf{s}, \mathbf{d}, \delta, \mathbf{w}) = \mathcal{B}(\mathbf{z}|\alpha_z, \beta_z)$, then

$$\mathcal{L}_z(\theta) = \mathbb{E}_{q_\theta(\mathbf{z}|\mathbf{s}, \mathbf{d}, \delta, \mathbf{w})} \left[ \text{KL}(q_\theta(\mathbf{z}|\mathbf{s}, \mathbf{d}, \delta, \mathbf{w})||p(\mathbf{z})) \right] \quad (54)$$

$$= \mathbb{E}_{q_\theta(\mathbf{z}|\mathbf{s}, \mathbf{d}, \delta, \mathbf{w})} \left[ \ln q_\theta(\mathbf{z}|\mathbf{s}, \mathbf{d}, \delta, \mathbf{w}) \right] + \mathbb{E}_{q_\theta(\mathbf{z}|\mathbf{s}, \mathbf{d}, \delta, \mathbf{w})} \left[ -\ln p(\mathbf{z}) \right] \quad (55)$$

$$= \sum_{i=0}^{hw-1} \left[ \ln \frac{\Gamma(\alpha_{z_i})\Gamma(\beta_{z_i})}{\Gamma(\alpha_{z_i} + \beta_{z_i})} + (\alpha_{z_i} - \alpha_z^0)\psi(\alpha_{z_i}) + (\beta_{z_i} - \beta_z^0)\psi(\beta_{z_i}) \right] \quad (56)$$

$$+ \sum_{i=0}^{hw-1} \left[ (\alpha_z^0 + \beta_z^0 - \alpha_{z_i} - \beta_{z_i})\psi(\alpha_{z_i} + \beta_{z_i}) \right] + C. \quad (57)$$

Here, $\Gamma(\cdot)$ and $\psi(\cdot)$ denote Gamma and Digamma functions, respectively. Suppose $q_\theta(\delta|\mathbf{s}, \mathbf{d}, \mathbf{w}) = \mathcal{N}(\delta|\mu_\delta, diag(\sigma_\delta^2))$, then

$$\mathcal{L}_\delta(\theta) = \mathbb{E}_{q_\theta(\delta|\mathbf{s}, \mathbf{d}, \mathbf{w})} \left[ \mathrm{KL}(q_\theta(\delta|\mathbf{s}, \mathbf{d}, \mathbf{w})||p(\delta)) \right] \tag{58}$$

$$= \mathbb{E}_{q_\theta(\delta|\mathbf{s}, \mathbf{d}, \mathbf{w})} \left[ \ln q_\theta(\delta|\mathbf{s}, \mathbf{d}, \mathbf{w}) \right] + \mathbb{E}_{q_\theta(\mathbf{z}|\mathbf{s}, \mathbf{d}, \delta, \mathbf{w})} \left[ -\ln p(\delta) \right] \tag{59}$$

$$= \frac{1}{2}\sigma_\delta^0 \|\mu_\delta\|_2^2 + \frac{1}{2}[\langle \sigma_\delta^0 \mathbf{1}, \sigma_\delta^2 \rangle - \langle \mathbf{1}, \ln \sigma_\delta^2 \rangle] + C. \tag{60}$$

Finally, suppose $q_\theta(\mathbf{w}|\mathbf{s}, \mathbf{d}) = \mathcal{N}(\mathbf{w}|\mu_w, diag(\sigma_w^2))$, then

$$\mathcal{L}_w(\theta) = \mathbb{E}_{q_\theta(\mathbf{w}|\mathbf{s}, \mathbf{d})} \left[ \mathrm{KL}(q_\theta(\mathbf{w}|\mathbf{s}, \mathbf{d})||p(\mathbf{w})) \right] \tag{61}$$

$$= \mathbb{E}_{q_\theta(\mathbf{w}|\mathbf{s}, \mathbf{d})} \left[ \ln q_\theta(\mathbf{w}|\mathbf{s}, \mathbf{d}) \right] + \mathbb{E}_{q_\theta(\mathbf{w}|\mathbf{s}, \mathbf{d})} \left[ -\ln p(\mathbf{w}) \right] \tag{62}$$

$$= \frac{1}{2}\sigma_w^0 \|\mu_w\|_2^2 + \frac{1}{2}[\langle \sigma_w^0 \mathbf{1}, \sigma_w^2 \rangle - \langle \mathbf{1}, \ln \sigma_w^2 \rangle] + C. \tag{63}$$

### A.2.5 Gaussian Splatting for Implicit Neural Representation

In this work, we sample 20 points on each edge and denote the entire graph with these points (including the sampled points and the nodes) as new nodes ($v$) and their connections as new edges ($e$). Then we input the new graph $\mathcal{G}(\mathcal{V}, \mathcal{E})$ into a Graph Neural Network to predict the parameters for Gaussian splatting, including the ($\sigma$, $\theta$, and $s$), where the feature of each node is its coordinates [46]. Note that the symbol definition is only used in this section for implicit neural representation explanation.

For the Gaussian splatting process, given a set of input nodes $\{\mathbf{v}_i\}_{i=1}^N$, where each node is associated with Gaussian parameters $(\sigma_i, \theta_i, s_i)$, the splatted field value at spatial location $\mathbf{u} \in \mathbb{R}^2$ is defined as:

$$f(\mathbf{u}) = \sum_{i=1}^N s_i \cdot \mathcal{N}(\mathbf{u} \mid \mathbf{v}_i, \Sigma_i)$$

where $\mathcal{N}(\mathbf{u} \mid \mathbf{v}_i, \Sigma_i)$ denotes a 2D anisotropic Gaussian:

$$\mathcal{N}(\mathbf{u} \mid \mathbf{v}_i, \Sigma_i) = \frac{1}{2\pi|\Sigma_i|^{1/2}} \exp\left( -\frac{1}{2}(\mathbf{u} - \mathbf{v}_i)^\top \Sigma_i^{-1}(\mathbf{u} - \mathbf{v}_i) \right)$$

The covariance matrix $\Sigma_i$ is constructed from the scale $\sigma_i$ and orientation $\theta_i$ as:

$$\Sigma_i = R(\theta_i) \begin{bmatrix} \sigma_i^2 & 0 \\ 0 & \epsilon \end{bmatrix} R(\theta_i)^\top \quad \text{with} \quad R(\theta_i) = \begin{bmatrix} \cos\theta_i & -\sin\theta_i \\ \sin\theta_i & \cos\theta_i \end{bmatrix}$$

where $\epsilon \ll \sigma_i^2$ is a small constant to ensure anisotropy.

For each pixel $u$ in the image grid, we can calculate the splatted field as $S^G$.

### A.2.6 Hyper-parameters

For Gaussian distributions $\mathcal{N}(\mu_w, \sigma_w)$ and $\mathcal{N}(\mu_\delta, \sigma_\delta)$, we set the prior distribution to be $\mathcal{N}(0, 1)$. For $\mathcal{N}(\mu_m, \sigma_m)$, we set the prior distribution to be $\mathcal{N}(0, 0.5)$ for the first 200 epochs for easier extraction of shape component, and then set it to be $\mathcal{N}(0, 1)$ for better decomposition of images. For the Gamma distributions, we set the prior distribution of $\mathcal{G}(\phi_\rho, \gamma_\rho)$ to be $\mathcal{G}(2, 10^{-6})$ and $\mathcal{G}(\phi_d, \gamma_d)$ to be $\mathcal{G}(2, 10^{-4})$. For the Beta distribution, we set the prior distribution of $\mathcal{B}(\alpha_z, \beta_z)$ to be $\mathcal{B}(2, 2)$. The sliding window size for unfolding is $k = 5$. For all losses used, we summed all items and divided them by $hw$.

## B  Experimental Details

### B.1  Evaluation Metrics

The evaluation metrics used in this paper are defined as follows:

Table 5: Cross-dataset evaluation: GraphSeg and BayeSeg are trained on CHASE and evaluated on CHASE, DRIVE, and HRF. U-Net is trained and tested on each dataset as a strong baseline. Numbers in parentheses indicate the performance drop relative to the results in Table 1. *Note that the same segmentation backbone was used for fair comparisons.*

| Test Set | Method | Accuracy | Soft Dice | F1 | Sensitivity | Specificity |
|---|---|---|---|---|---|---|
| CHASE | U-Net (train) | 95.17 | 79.74 | 78.39 | 81.97 | 96.79 |
| | BayeSeg (train) | 95.45 | 80.48 | 79.62 | 82.93 | 96.99 |
| | **GraphSeg (train)** | **96.54** | **82.91** | **81.82** | **87.84** | **97.40** |
| DRIVE | U-Net (train) | 90.89 | 61.09 | 61.09 | 48.55 | **98.51** |
| | BayeSeg | 92.34 ($\downarrow$2.14) | 71.68 ($\downarrow$7.40) | 70.71 ($\downarrow$7.85) | 61.85 ($\downarrow$15.75) | 97.87 ($\downarrow$0.81) |
| | **GraphSeg** | **93.56** ($\downarrow$2.57) | **77.16** ($\downarrow$8.07) | **76.01** ($\downarrow$8.81) | **68.09** ($\downarrow$14.93) | 98.19 ($\downarrow$0.04) |
| HRF | U-Net (train) | 89.79 | 71.37 | 67.76 | 78.15 | 91.74 |
| | BayeSeg | 92.05 ($\downarrow$3.30) | 70.73 ($\downarrow$11.04) | 70.13 ($\downarrow$10.92) | 67.71 ($\downarrow$7.73) | 96.13 ($\downarrow$2.29) |
| | **GraphSeg** | **93.38** ($\downarrow$2.50) | **78.16** ($\downarrow$6.13) | **76.59** ($\downarrow$7.30) | **78.90** ($\downarrow$2.68) | **96.88** ($\downarrow$1.19) |

- **Accuracy** = $\frac{TP+TN}{TP+TN+FP+FN}$,

- **F1** = $\frac{2 \cdot TP}{2 \cdot TP+FP+FN}$.

- **SoftDice** = $\frac{2 \sum_i y_i \hat{y}_i + \varepsilon}{\sum_i y_i^2 + \sum_i \hat{y}_i^2 + \varepsilon}$.

- **Sensitivity** = $\frac{TP}{TP+FN}$.

- **Specificity** = $\frac{TN}{TN+FP}$.

where TP, TN, FP, and FN denote true positive, true negative, false positive, and false negative pixels, respectively. $y$ and $\hat{y}$ denote the ground truth and prediction.

## B.2 Implementation details

We implement our method using PyTorch and train it on a cluster with NVIDIA A800 GPU. The batch size is set to 4, and the learning rate is set to 0.001. We use the Adam optimizer with a weight decay of 1e-5. The model is trained for 500 epochs, and we use early stopping based on the validation loss. The input images are resized to 256×256 pixels, and data augmentation techniques such as random rotation, and flipping are applied during training. For the image decomposition, we adopted the two ResNets with ten layers, and for the segmentation network, we used the efficient U-Net [35]. For the matching process, we use two two-layer graph convolutional networks (GCN) [47] with 64 channels and a three-layer convolutional network (CNN). We follow the training, validation and test data splitting in [11].

## B.3 Dataset Details

We employ four widely recognized datasets: CHASE [31], HRF [32], DRIVE [33], and STARE [34]. CHASE contains 28 color retina images with a size of 999×960 pixels. The HRF dataset consists of 45 color retina images with a size of 3504×2336 pixels. The DRIVE dataset contains 40 color retina images with a size of 584×565 pixels. The STARE dataset consists of 20 color retina images with a size of 700×605 pixels. In this work, we use the training set of one dataset and the test set of the other datasets to evaluate the generalization performance of our method. We also resize the images to the resolution of 256 × 256 pixels for training and testing.

## C  Additional Results

### C.1  Cross-domain Performance Drop

We also provide the cross-dataset relative performance drops of SOTA methods in table. 5. The noticeable performance drops demonstrates that there is still considerable room for improving the generalizability of current segmentation models.

Table 6: Computational cost comparison.

| Model | Parameters (M) | GFLOPs | Inference Time (s) | Memory (MB) |
|---|---|---|---|---|
| FSGNet | 18.32 | 89.59 | 6.23 | 925.42 |
| FRNet | 7.38 | 55.19 | 4.67 | 374.65 |
| AttUNet | 34.88 | 66.54 | 2.08 | 269.80 |
| AGNet | 9.33 | 16.73 | 5.99 | 330.72 |
| ConvUNeXt | 3.51 | 7.18 | 2.82 | 160.07 |
| DCSAU-net | 2.60 | 6.72 | 6.45 | 180.37 |
| R2UNet | 39.09 | 152.71 | 3.20 | 323.44 |
| SAUNet | 0.48 | 2.33 | 1.59 | 37.66 |
| BayeSeg | 19.32 | 88.82 | 7.95 | 144.50 |
| GraphSeg | 19.32 | 88.82 | 7.78 | 145.95 |

## C.2 Computational Complexity

We provide the computational cost comparison as follows. The computational costs are comparable for GraphSeg to other methods. In the inference stage, we only need to run the Decomposition (green in Figure 2) and Segmentation (blue in Figure 2) parts. Therefore, the computational costs are comparable. Besides, without the deformable graph prior, GraphSeg is downgraded to BayeSeg, thus the computational costs are almost the same for GraphSeg and BayeSeg for inference. We trained GraphSeg for 2.8 hours on GTX 4090. Since each method uses different training settings, it is not directly comparable. Therefore, we focus on inference metrics, which are more relevant for practice as shown in Table. 6.

## C.3 Results on Different Resolutions

Table 7: Comparison of GraphSeg performance with different image resolutions (128×128 vs. 256×256 nodes) on various datasets.

| Dataset | Resolution | Acc | Soft Dice | F1 | Sens | Spec |
|---|---|---|---|---|---|---|
| CHASE | 128 | 95.79 | 83.00 | 81.52 | 87.35 | 97.19 |
| | 256 | **96.54** | 82.91 | 81.82 | **87.84** | **97.40** |
| DRIVE | 128 | 95.19 | 84.00 | 83.50 | 80.45 | 97.88 |
| | 256 | **96.13** | **85.23** | **84.82** | **83.02** | **98.15** |
| HRF | 128 | 95.42 | 83.74 | 82.90 | 80.12 | 98.02 |
| | 256 | **95.88** | **84.29** | **83.89** | **81.58** | **98.07** |

We compare GraphSeg performance using two different image resolutions: 128×128 and 256×256. As shown in Table 7, increasing the image resolution generally improves segmentation quality across all datasets. In particular, the 256×256 pixel setting leads to consistent gains in accuracy and structure-sensitive metrics such as Dice and F1 on DRIVE and HRF. Notably, DRIVE benefits the most, with Dice increasing from 84.00 to 85.23 and F1 from 83.50 to 84.82. This suggests that higher-resolution images provide finer structural representation, enabling better modeling of small vessels and complex bifurcations. While the improvements are modest on CHASE, they remain stable, indicating the robustness of GraphSeg to image resolution.

## C.4 Visual Results

### C.4.1 Segmentation Results

We also present cross-domain visual results of segmentation to further demonstrate the generalization ability of **GraphSeg**. As shown in the visualizations, despite the significant structural variations across different datasets, **GraphSeg** consistently produces accurate and coherent vessel segmentation, highlighting its strong generalizability.

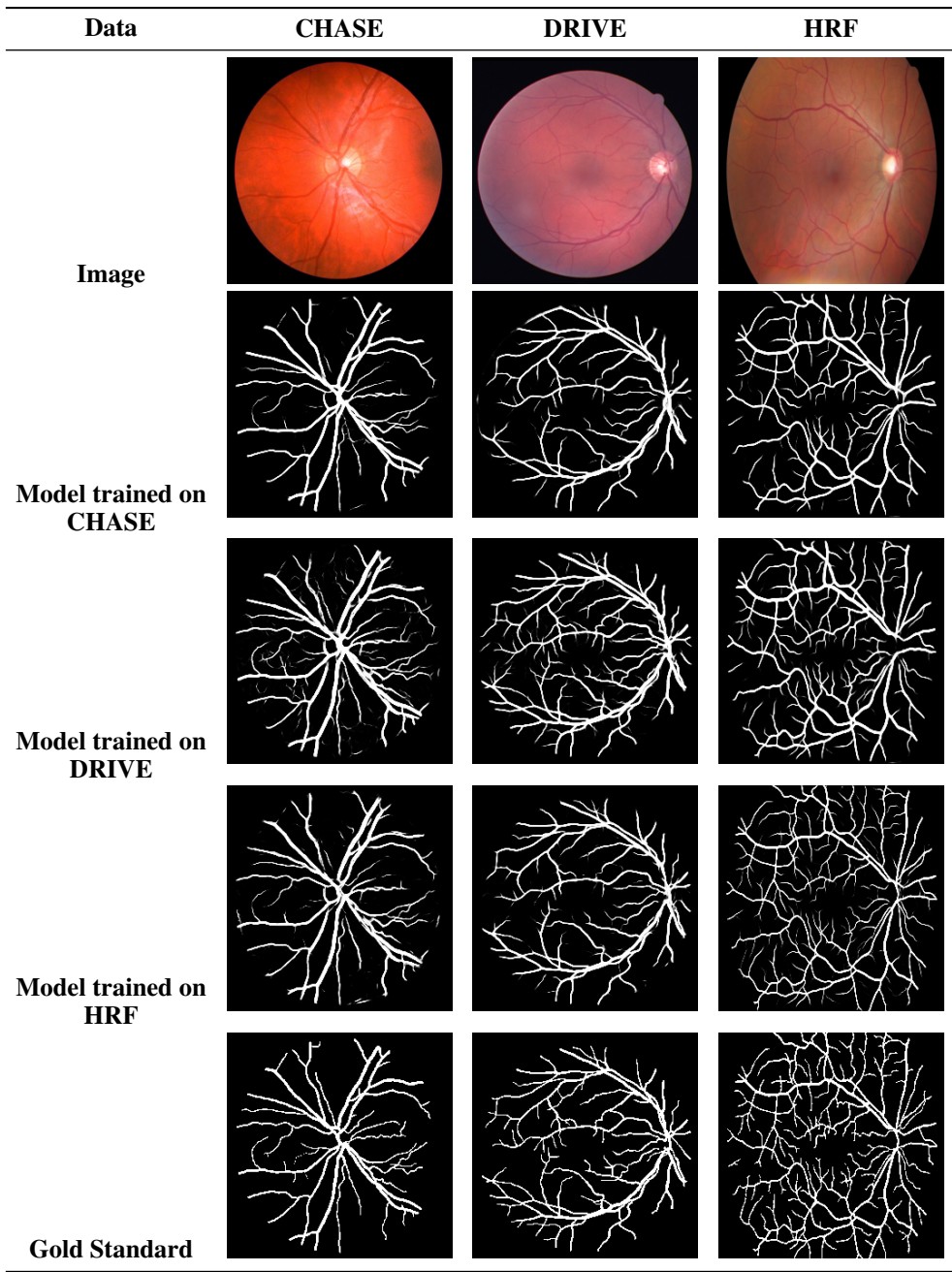

Figure 3: Cross-domain segmentation results from GraphSeg. Each column shows one dataset, while each row compares predictions from models trained on different domains.

### C.4.2 Decomposition Results

We further provide three groups of visual decomposition results in Fig. 4 to illustrate the effectiveness of our graph prior and image decomposition framework. To assess the cross-domain generalizability of **GraphSeg**, we additionally present cross-dataset visualizations in Fig. 5 and Fig. 6. As shown in these figures, the input images are consistently decomposed into a structure-preserved component ($s$) and a structure-degraded component ($m$). The segmentation is performed solely on the structure-preserved component $s$, which maintains a consistent style across different domains. This domain-invariant representation largely explains the strong generalization capability of **GraphSeg**.

| Image | $m$ | $s$ | $z$ |

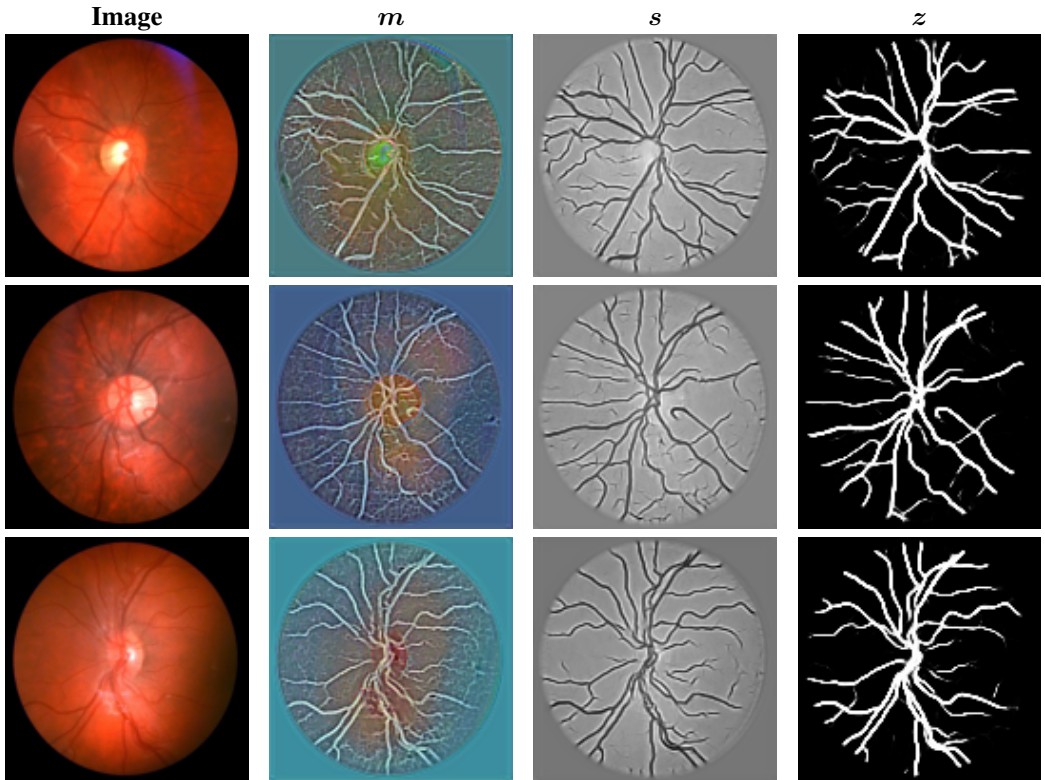

Figure 4: Visualization of image decomposition on CHASE dataset. Each row shows the input, structure-degraded component ($m$), structure-preserved component ($s$), and learned mask ($z$).

| Image | $m$ | $s$ | $z$ |

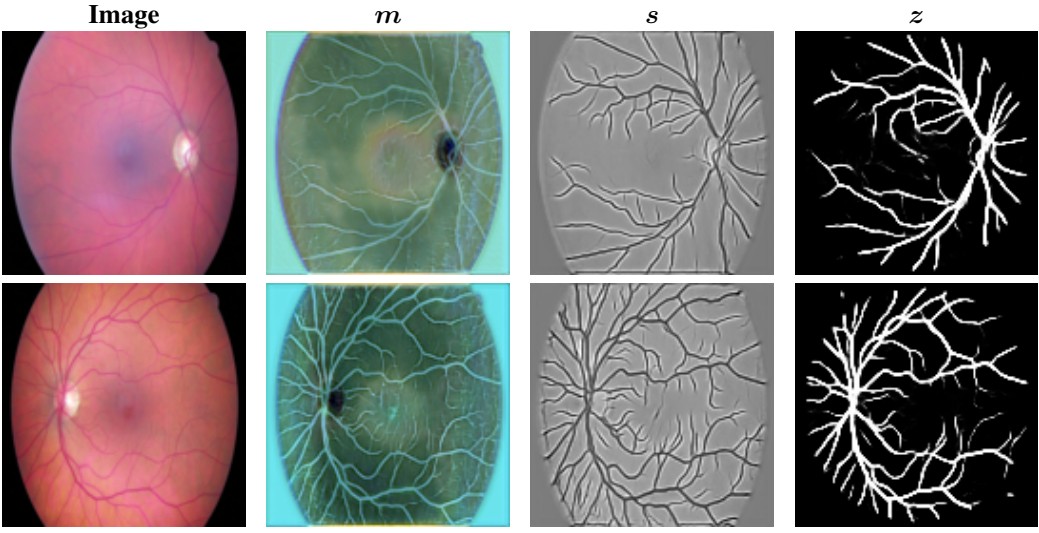

Figure 5: Cross dataset visualization of image decomposition on HRF dataset. GraphSeg is trained on CHASE dataset and tested on the other dataset. Each row shows the input, structure-degraded component ($m$), structure-preserved component ($s$), and learned mask ($z$).

### C.4.3 Sampled Vascular Structure for Training

We also provide illustrative examples in Fig. 7 to further explain the generalizability of GraphSeg. The sampled structure-preserved components ($s$) during training often contain substantial noise due to MCMC, which implicitly serves as a form of data augmentation. A model that can accurately segment

| **Image** | $m$ | $s$ | $z$ |
|:---:|:---:|:---:|:---:|

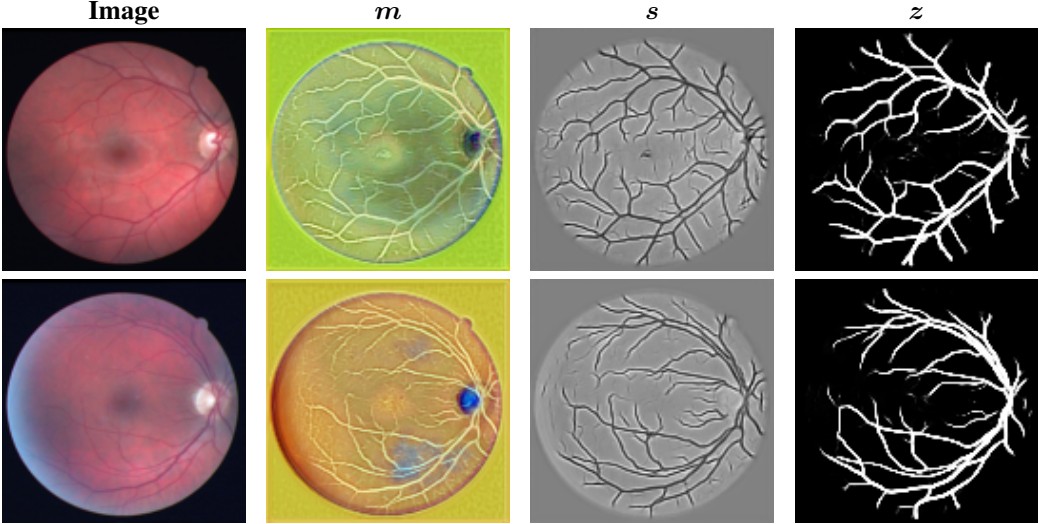

Figure 6: Cross dataset visualization of image decomposition on DRIVE dataset. GraphSeg is trained on CHASE dataset and tested on the other dataset. Each row shows the input, structure-degraded component ($m$), structure-preserved component ($s$), and learned mask ($z$).

the target structures from such noisy backgrounds demonstrates strong robustness and generalization capability.

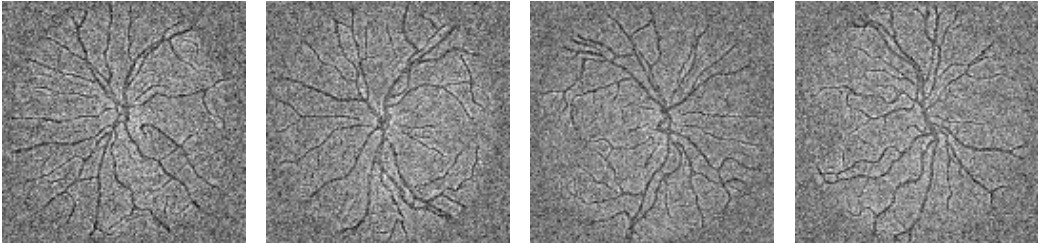

Figure 7: Sampled $s$ used for training.

