# OpenReview forum: "Towards Generalizable Retina Vessel Segmentation with Deformable Graph Priors"
_NeurIPS.cc/2025/Conference — NeurIPS 2025 poster_

### Official Review · Reviewer_D8Hv · 2025-06-30

**Clarity:** 2
**Significance:** 2
**Originality:** 3
**Rating:** 4
**Confidence:** 4

**Summary:**

The paper proposes GraphSeg, a novel variational Bayesian framework for retinal vessel segmentation that addresses the challenge of cross-domain generalization. GraphSeg builds upon Bayesian approaches such as BayeSeg and Indeed, and additionally integrates deformable graph priors derived from a statistical retinal atlas, enabling domain-invariant representations. A probabilistic graphical model (PGM) is introduced to jointly model images, vascular shapes, and graph priors, with a differentiable alignment mechanism guided by an unsupervised energy function. The framework leverages implicit neural representations and Gaussian splatting to align graph priors with image structures. Experiments on three public datasets—CHASE, DRIVE, and HRF—demonstrate GraphSeg’s superior performance, particularly in cross-domain settings, outperforming methods like the supervised trained U-Net, BayeSeg, and FSG-Net.

**Questions:**

1. Is GraphSeg the first method to use a retinal atlas or Bayesian framework for retinal vessel segmentation, or are there prior works that have explored similar anatomical priors? If so, how does GraphSeg differ from these approaches?
2. Given that the retinal atlas is derived from the CHASE dataset, how consistent is its distribution with those of other datasets like DRIVE and HRF, which may differ in imaging perspectives or conditions? Does the constructed retinal atlas generalize well across diverse datasets?
3. How does GraphSeg compare to FSG-Net, U-Net, and BayeSeg in terms of model parameters, computational cost (e.g., FLOPs), and training/testing speed? A quantitative comparison would help assess the practical deployability of GraphSeg.

**Ethical Concerns:**

["NO or VERY MINOR ethics concerns only"]

**Final Justification:**

After carefully considering the rebuttal and the discussion, I maintain a positive assessment of this work, for the following reasons:
1) The authors’ rebuttal provides additional experimental evidence that supports the effectiveness of the proposed method.
2) The use of a deformable statistical graph prior introduces a novel and original approach to addressing domain shift in retinal vessel segmentation.
3) One limitation is the reliance on preprocessing steps to extract vessel skeletons and construct graph representations. These steps may introduce variability and reduce the robustness and reproducibility of the overall pipeline.

**Limitations:**

yes

**Paper Formatting Concerns:**

No Formatting concern.

**Quality:**

3

**Strengths And Weaknesses:**

Strengths
1. Well-Motivated Research Problem. Retinal vessel segmentation is critical for diagnosing retinal diseases, but it faces challenges due to limited annotated data and significant domain variability across multi-center datasets. The paper effectively motivates the use of graph priors, specifically a statistical retinal atlas, to enhance cross-domain generalization.
2. Innovative Framework. The integration of a statistical retinal atlas as a deformable graph prior is a novel contribution. The use of retinal atlas derived from the CHASE dataset, combined with a variational Bayesian framework, introduces a fresh perspective to vessel segmentation. The ablation study in Table 3 convincingly demonstrates the effectiveness of the graph prior in improving generalization, with notable improvements in Soft Dice when trained on CHASE and tested on DRIVE.
3 . Comprehensive Methodology. GraphSeg combines multiple advanced techniques, including Bayesian modeling, graph construction with shape PCA, graph neural networks, and Gaussian splatting for implicit neural representation. The method's design is thorough, with a clear probabilistic graphical model (Fig. 1) and a detailed network architecture (Fig. 2). The use of an unsupervised energy function to guide structural alignment is particularly noteworthy, enabling the model to capture long-range vascular dependencies effectively.

Weaknesses
1. Insufficient Experimental Validation.
a) Preprocessing Dependency: the paper acknowledges the dependency on pre-processing for skeleton extraction and graph construction, which involve heuristic rules for junction merging and branch pruning. However, the extent of manual intervention in these steps is not clearly quantified, which could potentially lead to unfair comparisons with other methods. The authors should clarify the degree of human involvement and the robustness of these heuristics.
b) Limited Cross-Dataset Evaluation: while GraphSeg emphasizes strong cross-domain generalization, the evaluation is limited to three datasets (CHASE, DRIVE, HRF). To substantiate the claim of robust generalization, additional datasets such as STARE, DRHAGIS, LES-AV, and RETA should be included in Table 2 to cover a broader range of imaging conditions and vascular morphologies.
c) Limited Comparison Methods: The cross-dataset evaluation in Table 2 includes only U-Net and BayeSeg as comparison methods. More comparison methods are needed for Cross-dataset evaluation.
2. Unclear Presentation.
a) Vascular Structure Matching: The explanation of the vascular structure matching in Section 3.2 is not sufficiently clear. Specifically, the variable d in Equation (1) lacks a clear definition of its meaning and value range.
b) Comparison with BayeSeg: A more detailed comparison highlighting the technical distinctions and advantages would strengthen the contribution’s novelty.
c) Lack of Code Availability: Providing an anonymized code repository, including preprocessing steps, at submission would enhance trust in the results.

---

> ### Author Rebuttal · Authors · 2025-07-30
>
> Thank you for your constructive comments and kind support! All your concerns have been carefully addressed as below. The manuscript will be revised accordingly. We sincerely hope our responses fully address your questions.
> >> W1(a) Preprocessing Dependency: … the extent of manual intervention in these steps is not clearly quantified…
>
> A1(a): The graph atlas construction is only performed once on the CHASE training set before training. To evaluate the robustness, we use the graph prior from the CHASE to train GraphSeg on the DRIVE dataset. The results show: 1) GraphSeg(trained) outperforms UNet(trained) due to the graph prior, which demonstrates that the graph prior extracted from one dataset can be transferred to the other dataset. 2) GraphSeg outperforms UNet(trained) on most metrics, which confirms the generalizability of GraphSeg and demonstrates the robustness of GraphSeg to graph priors
>
> |Model|Acc|SoftDice|F1|Sen|Speci|
> |--|--|--|--|--|--|
> | ||**DRIVE(TRAIN)**| | | |
> |GraphSeg(trained)|96.13|85.23|84.82|83.02|98.15|
> |Unet(trained)|90.89|61.09|61.09|48.55|98.51|
> | | |**CHASE**| | | |
> |GraphSeg|95.02|82.56|79.37|89.70|95.69|
> |Unet(trained)|95.17|79.74|78.39|81.97|96.79|
> |GraphSeg(trained)|96.54|82.91|81.82|87.84|97.40|
> | | |**HRF**| | | |
> |GraphSeg|93.91|78.20|77.25|78.40|96.34|
> |Unet(trained)|89.79|71.37|67.76|78.15|91.74|
> |GraphSeg(trained)|95.88|84.29|83.89|81.58|98.07|
> >> W1(bc) Cross-Dataset Evaluation. … additional datasets and Comparison Methods…
>
> A1(bc): We provide additional results by including more datasets and additional seven methods, i.e., AttUNet[3], AGNet[4], ConvUNeXt[5], DCSAU-Net[6], R2UNet[7], SAUNet[8], CDARL[9]. FRSGNet[1] is current SOTA for retinal vessel segmentation
>
> GraphSeg outperforms other methods in generalization studies (on DRIVE and HRF), especially for F1 and Soft Dice. For other methods, since the F1 scores and soft **Dice scores are significantly lower, the acc, sensitivity and specificity are not very meaningful due to the significant class imbalance between the foreground (vessels) and background**
>
> The test results on the new STARE dataset show that GraphSeg outperforms other methods in soft Dice, F1, and Sensitivity, further confirming the superior generalizability of GraphSeg
>
> [1][ArXiv’25] Full-scale Representation Guided Network for Retinal Vessel Segmentation
>
> [2][BOE’24] a full-resolution dilated convolution network for retinal vessel segmentation
>
> [3][MIDL’18] Learning where to look for the pancreas
>
> [4][ISAIMS’20] Retinal Blood Vessel Segmentation via Attention Gate Network
>
> [5][KBS’22] An efficient convolution neural network for medical image segmentation
>
> [6][CBM’23] A deeper and more compact split-attention U-Net for medical image segmentation
>
> [7][JCAA’24] Recurrent residual convolutional neural network based on u-net (r2u-net) for medical image segmentation
>
> [8][MICCAI’20] Shape attentive u-net for interpretable medical image segmentation
>
> [9][MIA’24] Contrastive diffusion adversarial representation learning for label-free blood vessel segmentation
>
> |Model|Acc|SoftDice|F1|Sens|Speci|
> |--|--|--|--|--|--|
> | | |**CHASE(TRAIN)**| | | |
> |[1]|97.52|81.34|81.02|86.00|98.26|
> |[2]|97.26|80.10|79.10|84.74|97.65|
> |[3]|97.54|66.02|77.06|78.92|98.56|
> |[4]|97.44|76.25|77.58|85.14|98.13|
> |[5]|97.21|72.40|73.93|75.97|98.39|
> |[6]|97.17|74.33|74.68|79.87|98.12|
> |[7]|96.96|67.99|71.73|74.60|98.20|
> |[8]|97.47|76.17|77.23|82.02|98.33|
> |[9] (official)|78.53|11.54|0.52|0.84|83.60|
> |[9] (Reproduce)|83.18|16.97|15.69|26.05|86.93|
> |BayeSeg|95.45|80.48|79.62|82.93|96.99|
> |GraphSeg|96.54|82.91|81.82|87.84|97.40|
> | | |**DRIVE**| | | |
> |[1]|95.62|57.51|57.56|44.39|99.46|
> |[2]|96.65|71.33|73.57|67.53|98.85|
> |[3]|95.07|45.27|50.22|38.75|99.30|
> |[4]|96.25|69.46|70.59|65.21|98.59|
> |[5]|97.07|61.91|63.77|74.35|97.92|
> |[6]|95.30|62.30|62.56|56.82|98.19|
> |[7]|95.01|63.86|67.02|70.14|96.60|
> |[8]|96.46|69.78|70.67|61.92|99.06|
> |BayeSeg|92.34|71.68|70.71|61.85|97.87|
> |GraphSeg|93.56|77.16|76.01|68.09|98.19|
> | | |**HRF**| | | |
> |[1]|95.90|68.26|68.56|71.50|98.28|
> |[2]|97.00|64.99|67.56|69.78|97.28|
> |[3]|97.37|54.20|65.29|66.41|97.94|
> |[4]|96.70|63.03|64.54|66.69|97.10|
> |[5]|95.62|59.74|60.60|49.56|99.07|
> |[6]|96.63|61.70|62.09|68.66|97.30|
> |[7]|94.21|49.07|52.03|88.87|94.40|
> |[8]|97.15|65.87|67.17|83.92|97.65|
> |BayeSeg|92.05|70.73|70.13|67.71|96.13|
> |GraphSeg|93.38|78.16|76.59|78.90|96.88|
> |||**STARE**||||
> |[1]|96.15|54.08|54.23|45.43|99.33|
> |[2]|93.04|64.91|67.07|62.39|98.96|
> |[3]|95.71|37.91|41.97|34.39|99.54|
> |[4]|96.52|67.49|68.82|66.74|98.47|
> |[5]|95.97|52.65|53.30|46.89|98.96|
> |[6]|94.77|47.59|47.83|45.65|97.75|
> |[7]|95.53|49.90|50.86|50.63|98.22|
> |[8]|96.87|61.50|62.23|56.44|99.29|
> |BayeSeg|92.85|67.82|66.82|59.78|97.65|
> |GraphSeg|93.65|72.76|71.43|68.72|97.69|
> >> W2(a) Definition of the variable $d$ in Eq (1).
>
> A2(a): The variable $d$ represents the landmark detection variable that identifies the points where vascular structures exhibit significant geometric dissimilarity. These are typically the branching points or bifurcations of blood vessels, which is firstly introduced in line 153
>
> The variable $d$ follows a Gamma prior distribution, and its value is defined within the context of the model’s optimization to help guide the detection of landmarks by enforcing structural constraints during the vascular structure matching
> >> W2(b) Comparison with BayeSeg: A more detailed comparison would strengthen the contribution’s novelty
>
> A2(b): The key technical differences between GraphSeg and BayeSeg as follows:
> - Graph-based Prior: Comparing to BayeSeg, GraphSeg introduces a deformable retinal graph prior that allows for better modeling of vascular structures with topological flexibility. The deformable graph prior, adapting to different vascular morphologies, is a key distinction that enhances the robustness and generalizability of GraphSeg across domains, as demonstrated in the ablation study
> - Anatomical Consistency: BayeSeg focuses on shape-based decomposition without explicit structural alignment to a prior, while GraphSeg integrates anatomical constraints through the deformable graph prior, leading to better vessel structure continuity
> - Domain Generalization: By jointly modeling anatomical topology and image structure, GraphSeg performs better in unseen domains, as evidenced by experiments on multiple retinal datasets
>
> Notably, the deformable graph priors are derived from a shared retinal atlas, enabling the modeling of dependencies between vascular structures across different retinal images. This fundamentally distinguishes GraphSeg from conventional models (e.g., BayeSeg), which operate independently on each image under the i.i.d. assumption, thus neglecting cross-sample anatomical dependencies.
> >> W2(c) Lack of Code Availability.
>
> A2(c): We are committed to making the entire code publicly available upon acceptance of the paper. The pre-processing strictly follows [1] by performing their code on the CHASE dataset.
>
> [1] PAMI’24. Statistical analysis of complex shape graphs.
> >> Q3. Is GraphSeg the first method to use a retinal atlas or Bayesian framework for retinal vessel segmentation? how does GraphSeg differ from others?
>
> A3. To our knowledge, GraphSeg is the first method to combine a deformable graph prior with a Bayesian framework for retinal vessel segmentation. While prior works have explored Bayesian frameworks and anatomical priors separately, GraphSeg introduces a novel approach by using deformable retinal graph priors that adapt to various vascular structures across datasets. This is a unique contribution that distinguishes GraphSeg from previous methods.
>
> Previous works like BayeSeg have employed Bayesian frameworks for segmentation but only focused on compact anatomies such as heart and prostate. Similarly, graph-based approaches have been used in CT vessel segmentation [1], but they tried to construct image-specific graphs from the first-stage segmentations to further refine segmentations by graph convolutional networks, which is different with our image-agnostic deformable graph priors constructed from retinal Atlas for informing vessel topology.
>
> [1] Graph Convolution Based Cross-Network Multi-Scale Feature Fusion for Deep Vessel Segmentation
> >> Q4. ...how consistent is the distribution of the constructed retinal atlas... the generalization of the constructed retinal atlas...
>
> A4. The distribution of vessel graphs are different as shown in Figure 3 in appendix. However, since our graphs are deformable with $\delta$ for capturing displacement, the graph atlas extracted from CHASE can be transferred to other datasets. To evaluate the generalizability, we use the graph prior from the CHASE to train GraphSeg on the DRIVE dataset. The results and explanation are shown in the response A1(a) to W1(a). The results show that the graph prior extracted from one dataset can actually be used for the other dataset.
> >> Q5. ...computational complexity...
>
> A5. We provide the computational cost comparison as follows. The computational costs are comparable for GraphSeg to other methods. In the inference stage, we only need to run the Decomposition (green in Figure 2) and Segmentation (blue in Figure 2) parts. Therefore, the computational costs are comparable. Besides, without the Deformable graph prior, GraphSeg is downgraded to BayeSeg, thus the computational costs are almost the same for GraphSeg and BayeSeg for inference. We trained GraphSeg for 2.8 hours on GTX 4090. Since each method uses different training settings, it is not directly comparable. Therefore, we focus on inference metrics which are more relevant for practice.
>
> |Model|Param(M)|GFLOPs|InferTime(s)|Mem(MB)|
> |--|--|--|--|--|
> |[1]|18.32|89.59|6.23|925.42|
> |[2]|7.38|55.19|4.67|374.65|
> |[3]|34.88|66.54|2.08|269.80|
> |[4]|9.33|16.73|5.99|330.72|
> |[5]|3.51|7.18|2.82|160.07|
> |[6]|2.60|6.72|6.45|180.37|
> |[7]|39.09|152.71|3.20|323.44|
> |[8]|0.48|2.33|1.59|37.66|
> |BayeSeg|19.32|88.82|7.95|144.50|
> |GraphSeg|19.32|88.82|7.78|145.95|

---

> > ### Comment · Reviewer_D8Hv · 2025-08-04
> >
> > I would like to thank the authors for their detailed response to all my questions. The additional results — including the comparison methods, cross-dataset evaluation, and computational cost — are very helpful, and I hope these can be incorporated into the updated manuscript.

---

> > > ### Author Response · Authors · 2025-08-04
> > > **Thanks for Your Positive Feedback**
> > >
> > > We sincerely thank the reviewer for the positive feedback. The suggested additional results (including comparison methods, cross-dataset evaluation, and computational cost) will be incorporated into the revised manuscript as recommended.

---

### Official Review · Reviewer_EXz9 · 2025-07-01

**Clarity:** 4
**Significance:** 3
**Originality:** 3
**Rating:** 4
**Confidence:** 5

**Summary:**

The authors propose GraphSeg, a variation of Bayesian framework.
They introduce a deformable graph prior module in training this network and design a statistical model to align the vasculature and deformable graph.
Their deisgn demonstrates superior performance comparing to SOTA while remaining good generalizability.

**Questions:**

1. Graph Atlas is mentioned four times in the paper. Could you please clarify how the retinal vascular graph atlas is initially obtained?
2. In the cross-dataset evaluation, it is unclear whether pre-processing and decomposition were applied consistently across all compared methods, or only to the proposed approach. Could you provide a cross-dataset comparison regarding pre-processing? Pre-processing can significantly affect the assessment of generalization ability, and its impact should be clearly demonstrated.
3. The datasets used in the paper are relatively small and dated (a total of 28+40+45=113 images from 2004, 2013, and 2013). To better validate the performance and robustness of the proposed model, it would be important to include more recent and larger-scale datasets.
4. In Table 3, sensitivity increases significantly with the addition of more components. Could you explain the reason behind this?

I will raise Originality if 4 and 3 are accomodated; will raise Quality if 1,2,3 are solved.

**Ethical Concerns:**

["NO or VERY MINOR ethics concerns only"]

**Final Justification:**

Thank you for your reply! I recognize the authors fully address my concerns on Originality and partially address my concerns on Quality. I will raise my score.
My remaining concern is that the preprocessing may inadvertently align the data distribution in a way that favors the proposed model, potentially leading to an overestimation of its generalization ability, and that the dataset aspect (inclusion of more datasets and having a detailed introduction on the dataset usage) may affect reproducibility and the assessment of generalization in broader scenarios.
This is the evaluation concern that draws the Borderline accept.

**Limitations:**

yes

**Paper Formatting Concerns:**

Not found

**Quality:**

2

**Strengths And Weaknesses:**

Strengths:
1. Clarity: The writing is clear and easy to follow.
2. Significance:
 - Strong segmentation performance on classic benchmarks.
 - Generalization is a practical challenge in medical imaging. The authors explore the potential of Bayesian frameworks to address this issue.
3. Originality:
The use of retinal priors, such as landmark detection, reflects a solid integration with real-world clinical scenarios. I personally appreciate this practical approach.

Weaknesses:
1. Quality:
 - The experiments may not fully support the claimed superiority of the method. There is a lack of repeated trials to assess consistency and robustness.
 - Scale of Data: The dataset used is relatively small, which may limit the generalizability of the findings.
 - The cross-dataset generalization experiments are limited and not comprehensive enough to confirm the method’s robustness.

2. Originality: The limitation in experiments makes it difficult to judge the reliability and practical value of the method. Since the paper use many exisiting ideas without novel changes such as landmark detection, Bayesian modeling, or image decomposition, the paper should at least provide strong experimental evidence (not just in discussion) to show the advantage of combining these ideas (why not the others) or highlight some design principles of the framework. These may help the paper demonstarate the uniqueness of their pipeline.

---

> ### Author Rebuttal · Authors · 2025-07-30
>
> Thank you for your comments! All your concerns have been carefully addressed as below. The manuscript will be carefully revised accordingly. We sincerely hope our responses fully address your questions.
>
> >> W1.1 The experiments may not fully support the claimed superiority of the method. There is a lack of repeated trials to assess consistency and robustness.
>
> A1.1. Thanks for your comment. The improvement of GraphSeg in generalizability study is significant. We also perform a statistical test as follows. GraphSeg outperforms BayeSeg significantly as shown in the table
>
> - CHASE DB
>
> | Metric| t-test   | Wilcoxon | Significant (p < 0.05) |
> |-----|----------|----------|------------------------|
> | Accuracy    | 0  | 0.00012  | TRUE |
> | Soft Dice   | 0.00001  | 0.00024  | TRUE |
> | F1    | 0  | 0.00012  | TRUE |
> | Sensitivity | 0.00078  | 0.00037  | TRUE |
> | Specificity | 0.00022  | 0.00061  | TRUE |
>
> - DRIVE
>
> | Metric| t-test   | Wilcoxon | Significant (p < 0.05) |
> |-------------|----------|----------|------------------------|
> | Accuracy    | 0.00004  | 0.00021  | TRUE |
> | Soft Dice   | 0.00483  | 0.00639  | TRUE |
> | F1    | 0.00013  | 0.00059  | TRUE |
> | Sensitivity | 0.00554  | 0.00422  | TRUE |
> | Specificity | 0.00016  | 0.00059  | TRUE |
>
> - HRF
>
> | Metric| t-test   | Wilcoxon | Significant (p < 0.05) |
> |-------------|----------|----------|------------------------|
> | Accuracy    | 0.00018  | 0.00002  | TRUE |
> | Soft Dice   | 0.00004  | 0  | TRUE |
> | F1    | 0  | 0  | TRUE |
> | Sensitivity | 0  | 0  | TRUE |
> | Specificity | 0.00502  | 0.00248  | TRUE |
>
>
>
> >> W.1.2 Scale of Data: The dataset used is relatively small, which may limit the generalizability of the findings. The cross-dataset generalization experiments are limited and not comprehensive enough to confirm the method’s robustness.
>
> A1.2. Thank you for your comment. The three datasets used in this work (DRIVE, HRF, and CHASE) are widely recognized as standard benchmarks for retinal vessel segmentation [1]. To further evaluate the generalizability of GraphSeg, we have also conducted experiments on the STARE dataset and compared with seven additional methods, including AttUNet [6], AGNet [7], ConvUNeXt [2], DCSAU-Net [3], R2UNet [4], SAUNet [8], and CDARL [9]. All models were trained on the CHASE dataset, and evaluated by ACC, Soft Dice, F1 score, Sensitivity, and Specificity. **Note that Soft Dice, F1 score, Sensitivity are more appropriate** to evaluate vessel segmentation performance *due to the significant class imbalance between the foreground (vessels) and background*. From the test results on STARE one can see that GraphSeg outperforms other methods in soft Dice, F1 score, and Sensitivity, further confirming the superior generalizability of GraphSeg.
>
> - CHASE (train)
>
> | Model   | Acc    | Soft Dice | F1    | Sensitivity | Specificity |
> |----------------|--------|-----------|-------|-------------|-------------|
> | FSGNet  | 97.52  | 81.34     | 81.02 | 86.00| 98.26|
> | FRNet   | 97.26  | 80.10     | 79.10 | 84.74| 97.65|
> | AttUNet | 97.54  | 66.02     | 77.06 | 78.92| 98.56|
> | AGNet   | 97.44  | 76.25     | 77.58 | 85.14| 98.13|
> | ConvUNeXt      | 97.21  | 72.40     | 73.93 | 75.97| 98.39|
> | DCSAU-net     | 97.17  | 74.33     | 74.68 | 79.87| 98.12|
> | R2UNet  | 96.96  | 67.99     | 71.73 | 74.60| 98.20|
> | SAUNet  | 97.47  | 76.17     | 77.23 | 82.02| 98.33|
> | CDARL (official) | 78.53 | 11.54    | 0.52  | 0.84 | 83.60|
> | CDARL (Reproduce) | 83.18 | 16.97  | 15.69 | 26.05| 86.93|
> | BayeSeg | 95.45  | 80.48     | 79.62 | 82.93| 96.99|
> | GraphSeg| 96.54|82.91|81.82|87.84|97.40|
>
> - DRIVE
>
> | Model   | Acc    | Soft Dice | F1    | Sensitivity | Specificity |
> |----------------|--------|-----------|-------|-------------|-------------|
> | FSGNet  | 95.62  | 57.51     | 57.56 | 44.39| 99.46|
> | FRNet   | 96.65  | 71.33     | 73.57 | 67.53| 98.85|
> | AttUNet | 95.07  | 45.27     | 50.22 | 38.75| 99.30|
> | AGNet   | 96.25  | 69.46     | 70.59 | 65.21| 98.59|
> | ConvUNeXt      | 97.07  | 61.91     | 63.77 | 74.35| 97.92|
> | DCSAU-net     | 95.30  | 62.30     | 62.56 | 56.82| 98.19|
> | R2UNet  | 95.01  | 63.86     | 67.02 | 70.14| 96.60|
> | SAUNet  | 96.46  | 69.78     | 70.67 | 61.92| 99.06|
> | BayeSeg | 92.34  | 71.68     | 70.71 | 61.85| 97.87|
> | GraphSeg| 93.56  | 77.16     | 76.01 | 68.09| 98.19|
>
> - HRF
>
> | Model   | Acc    | Soft Dice | F1    | Sensitivity | Specificity |
> |----------------|--------|-----------|-------|-------------|-------------|
> | FSGNet  | 95.90  | 68.26     | 68.56 | 71.50| 98.28|
> | FRNet   | 97.00  | 64.99     | 67.56 | 69.78| 97.28|
> | AttUNet | 97.37  | 54.20     | 65.29 | 66.41| 97.94|
> | AGNet   | 96.70  | 63.03     | 64.54 | 66.69| 97.10|
> | ConvUNeXt      | 95.62  | 59.74     | 60.60 | 49.56| 99.07|
> | DCSAU-net     | 96.63  | 61.70     | 62.09 | 68.66| 97.30|
> | R2UNet  | 94.21  | 49.07     | 52.03 | 88.87| 94.40|
> | SAUNet  | 97.15  | 65.87     | 67.17 | 83.92| 97.65|
> | BayeSeg | 92.05  | 70.73     | 70.13 | 67.71| 96.13|
> | GraphSeg| 93.38|78.16|76.59|78.90|96.88|
>
> - STARE
>
> | Model   | Acc    | Soft Dice | F1    | Sensitivity | Specificity |
> |----------------|--------|-----------|-------|-------------|-------------|
> | FSGNet  | 96.15  | 54.08     | 54.23 | 45.43| 99.33|
> | FRNet   | 93.04  | 64.91     | 67.07 | 62.39| 98.96|
> | AttUNet | 95.71  | 37.91     | 41.97 | 34.39| 99.54|
> | AGNet   | 96.52  | 67.49     | 68.82 | 66.74| 98.47|
> | ConvUNeXt      | 95.97  | 52.65     | 53.30 | 46.89| 98.96|
> | DCSAU-net     | 94.77  | 47.59     | 47.83 | 45.65| 97.75|
> | R2UNet  | 95.53  | 49.90     | 50.86 | 50.63| 98.22|
> | SAUNet  | 96.87  | 61.50     | 62.23 | 56.44| 99.29|
> | BayeSeg | 92.85  | 67.82     | 66.82 | 59.78| 97.65|
> | GraphSeg| 93.65  | 72.76     | 71.43 | 68.72| 97.69|
>
> [1] [ArXiv’25] FSG-Net: Full-scale Representation Guided Network for Retinal Vessel Segmentation
>
> [2] [Knowledge-based systems’22] ConvUNeXt: An efficient convolution neural network for medical image segmentation
>
> [3] [Computers in Biology and Medicine’23] DCSAU-Net: A deeper and more compact split-attention U-Net for medical image segmentation
>
> [4] [Journal of Computational Analysis & Applications’24] Recurrent residual convolutional neural network based on u-net (r2u-net) for medical image segmentation
>
> [5] [Biomedical Optics Express’24] FRD-Net: a full-resolution dilated convolution network for retinal vessel segmentation
>
> [6] [Medical Imaging with Deep Learning’18] Attention u-net: Learning where to look for the pancreas
>
> [7] [ISAIMS’20] AGNet：Retinal Blood Vessel Segmentation via Attention Gate Network
>
> [8] [MICCAI’20] Saunet: Shape attentive u-net for interpretable medical image segmentation
>
> [9] [MedIA'24] C-DARL: Contrastive diffusion adversarial representation learning for label-free blood vessel segmentation
>
> >> W2. Originality… The limitation in experiments makes it difficult to judge the reliability and practical value of the method…
>
> A2. Thanks for your concern. We provide additional results as shown in the table above. Experiments show the superiority of our GraphSeg. We would like to clarify the originality of our contributions. Specifically, our work introduces: (1) a novel Bayesian retinal vessel segmentation framework that goes beyond traditional methods focusing on compact shapes; (2) the development of deformable retinal graph priors for aligning heterogeneous vascular structures in retinal images, along with a statistical model to drive the alignment between image vascular structures and these graph priors; and (3) extensive validation of our framework across multiple widely used datasets, demonstrating its superior generalizability in unseen scenarios.
>
> >> Q3. Could you please clarify how the retinal vascular graph atlas is initially obtained?
>
> A3. Thanks for your concern. We strictly follow [1] to construct the graph Atlas as we described in Appendix.A.1
>
> [1] PAMI’24. Statistical analysis of complex shape graphs.
>
> >>Q4. …whether pre-processing and decomposition were applied consistently across all compared methods…
>
> A4. Thank you for your valuable comment. In our cross-dataset evaluation, we ensured consistency in the pre-processing steps across all compared methods to provide a fair comparison. Specifically, the same pre-processing pipeline (including image resizing, normalization, and augmentation) was applied uniformly to all methods, including the proposed GraphSeg approach.
>
> For the decomposition step, which is specific to GraphSeg and BayeSeg, we only applied it to these two methods and not to the baseline approaches. We will clarify this point in the revised manuscript and emphasize that while pre-processing was consistent, the combination of decomposition and deformable graph priors were unique to GraphSeg. It’s worth to mention that the decomposition and deformable graph priors in GraphSeg are trained end-to-end instead of stage-by-stage.
>
> >> Q5. … To better validate the performance and robustness of the proposed model, it would be important to include more recent and larger-scale datasets….
>
> A5. Please refer to A1.2.
>
> >> Q6. In Table 3, sensitivity increases significantly with the addition of more components. Could you explain the reason behind this?
>
> A6. Thank you for your insightful question. The significant increase in sensitivity observed with the addition of more components in Table 3 is attributed to the model’s ability to capture finer details of the vascular structures, particularly in challenging cases where small or faint vessel segments are present.
>
> As more components are introduced, the model becomes better at distinguishing between foreground vascular structures and the background, improving its sensitivity to vessels that might otherwise be overlooked. This is especially beneficial in scenarios with complex or noisy images, where the model needs to identify subtle vascular features.

---

> ### Comment · Reviewer_EXz9 · 2025-08-05
>
> Thank you for your reply! I recognize the authors fully address my concerns on Originality and partially address my concerns on Quality. I will raise my score, and I would appreciate the authors considering the following suggestions.
>
> By pointing out Q1 (Q3 in rebuttal with authors' answer A.3), I was expecting there would be a specific design for retina images, as the retina is of a different imaging domain from conventional graphics, and the authors claim this as their first contribution. I understand the authors did make some adjustments to the Graph Atlas, but these parameter-level adjustments are still under the original scope of the existing Graph Atlas method. However, I still appreciate this effort and encourage the authors to include the corresponding paragraph in the main paper, as it is an important component.
>
> For the Q2 (Q4 in rebuttal with answers A4), I have queried from the "evaluation" perspective, to make sure pre-processing will help evaluate generalisability rather than hinder it. Specifically, if the preprocessing aligns the input distribution more closely with the domain assumptions of the proposed model, it can lead to artificially improved generalization performance compared to other methods. This could misrepresent the true robustness of the approach in out-of-domain settings. I strongly encourage the authors to describe their preprocessing pipeline in greater detail. It is important, as the authors claimed generalization in the title and the third contribution.
>
> For Q3 (Q5 in rebuttal with answers A5 and A1.2), STARE is also a widely used dataset. It initially had very few images, eventually reaching 400 images. The authors should clearly specify which version or how much data they used in their experiments. I understand the data scarcity and will not insist that authors include more datasets. However, I still encourage the authors to test their results on additional datasets to evaluate generalization beyond segmentation. If the authors' experiments cover a wider dataset for generalization, I may further raise the score.
>
> For Q4 (Q6 in rebuttal with A6), I recommend further discussion in the paper, as the analysis of experimental results is necessary.

---

### Official Review · Reviewer_pSst · 2025-07-01

**Clarity:** 3
**Significance:** 3
**Originality:** 3
**Rating:** 5
**Confidence:** 3

**Summary:**

The paper introduces a novel method for segmenting vascular structures in color fundus photographs.  The main novelty of the method is to use a deformable graph prior for the vascular network within a Bayesian segmentation framework.  Results show modest improvements against a reasonable set of baselines on several data sets from different sources.  However, the strongest result is in showing greater improvements against baselines in cross-data-set tasks, i.e. training on one and testing on the other.  This suggests greater robustness and generalizability of the proposed approach.

**Questions:**

Does the presence of pathology affect performance?  An algorithm that works on normal cases (the majority) but fails in the presence of pathology has little clinical utility.

**Ethical Concerns:**

["NO or VERY MINOR ethics concerns only"]

**Final Justification:**

No update.

**Limitations:**

Yes

**Quality:**

3

**Strengths And Weaknesses:**

Strengths

This is an important problem with clear clinical utility.

The methods appear sound and well thought out.

The experiments and clear and the results nicely demonstrate potential benefit.

Weaknesses

Does not distinguish veins and arteries unlike other recent work of Zhou et al (in particular https://www.sciencedirect.com/science/article/abs/pii/S1361841524000239 but also https://tvst.arvojournals.org/article.aspx?articleid=2783477), which should be considered in the related work and discussion.

Does not consider the effect of pathology in the experimental evaluation.

---

> ### Author Rebuttal · Authors · 2025-07-30
>
> Thank you for your constructive comments and kind support! All your concerns have been carefully addressed as below. The manuscript will be carefully revised accordingly. We sincerely hope our responses fully address your questions.
>
> >> W1. Does not distinguish veins and arteries
>
> A1. We acknowledge the importance of distinguishing between veins and arteries in retinal vessel segmentation, especially for clinical applications. While our approach focuses on general vascular segmentation, we recognize that specialized methods differentiating veins from arteries, such as those proposed by Zhou et al., may offer additional clinical value. We appreciate the reviewer pointing out this gap, and we will include a discussion on this in the revised manuscript, highlighting the potential for extending our method to vein/artery differentiation. We will also reference Zhou et al.’s work in the related work section to clarify how their method differs from ours and discuss the trade-offs between our generalizable approach and more specialized methods.
>
> >> W2&Q3. Does not consider the effect of pathology in the experimental evaluation. Does the presence of pathology affect performance? An algorithm that works on normal cases (the majority) but fails in the presence of pathology has little clinical utility.
>
> A2. Thank you for your insightful comment. **The HRF dataset includes both healthy and pathological images (e.g., diabetic retinopathy)**, and the DRIVE and CHASE datasets predominantly contain healthy images, with only minor pathological cases. GraphSeg performs well across all three datasets, particularly with respect to generalizability. Although our current experiments focus on generalization across different retinal datasets, we agree that pathology could significantly impact segmentation performance.

---

> > ### Comment · Reviewer_pSst · 2025-08-05
> > **Brief response**
> >
> > Thank you for your answers.  I stand by my original assessment.

---

### Official Review · Reviewer_TdpE · 2025-07-03

**Clarity:** 3
**Significance:** 4
**Originality:** 4
**Rating:** 4
**Confidence:** 4

**Summary:**

This paper proposes GraphSeg, a method that addresses cross-domain generalization in retinal vessel segmentation by combining deformable graph priors with structure-aware image decomposition. Experiments on three public datasets (CHASE, DRIVE, HRF) demonstrate consistent improvements over existing methods under domain shift, validating the importance of jointly modeling anatomical topology and image structure.

**Questions:**

1. The paper lacks a comprehensive analysis of the computational overhead introduced by the variational Bayesian framework and deformable graph alignment.
2. How sensitive is the method to the quality and representativeness of the statistical retinal atlas used for graph prior construction?
3. The paper appears to lack a comprehensive comparison with recent end-to-end deep learning approaches for vessel segmentation (e.g., U-Net variants, Transformer-based methods, recent CNN architectures).

**Ethical Concerns:**

["NO or VERY MINOR ethics concerns only"]

**Limitations:**

The authors acknowledge the dependency on preprocessing steps for skeleton extraction and graph construction, which represents a major technical limitation. However, the paper also lacks computational complexity discussion, comparison with state-of-the-art end-to-end methods, and failure case analysis specific to the proposed approach.

**Quality:**

4

**Strengths And Weaknesses:**

Strengths:
1. The integration of deformable graph priors into a Bayesian segmentation framework represents a significant methodological advancement beyond conventional shape-based priors, specifically addressing the complex vascular morphology challenges.
2. The method demonstrates superior performance on unseen scenarios across multiple datasets, addressing a critical limitation in medical image analysis where domain adaptation is essential for clinical deployment.

Weakness:
1. The method requires preprocessing steps to extract vessel skeletons from segmentation masks and construct graph representations. This introduces potential variability due to heuristic morphological operations (e.g., junction merging, branch pruning) and limits the end-to-end learning capability of the framework.
2. The combination of variational Bayesian inference with deformable graph alignment may introduce significant computational overhead compared to standard CNN-based approaches. The paper lacks a discussion of runtime complexity and scalability to higher image resolutions.

---

> ### Author Rebuttal · Authors · 2025-07-30
>
> Thank you for your constructive comments and kind support! All your concerns have been carefully addressed as below. The manuscript will be carefully revised accordingly. We sincerely hope our responses fully address your questions.
>
> >> Q1. The paper lacks a comprehensive analysis of the computational overhead introduced by the variational Bayesian framework and deformable graph alignment.
>
> A1. Thanks for your suggestion, we provide the computational cost comparison as follows. The computational costs are comparable for GraphSeg to other methods. In the inference stage, we only need to run the Decomposition (green in Figure 2) and Segmentation (blue in Figure 2) parts. Therefore, the computational costs are comparable. Besides, without the Deformable graph prior, GraphSeg is downgraded to BayeSeg, thus the computational costs are almost the same for GraphSeg and BayeSeg for inference. We trained GraphSeg for 2.8 hours on GTX 4090. Since each method uses different training settings, it is not directly comparable. Therefore, we focus on inference metrics, which are more relevant for practice.
>
> | Model| Parameters | GFLOPs | Inference Time (s) | Memory (MB) |
> |-------------|------------|--------|--------------------|-------------|
> | FSGNet      | 18.32 M    | 89.59  | 6.23 | 925.42      |
> | FRNet| 7.38 M     | 55.19  | 4.67 | 374.65      |
> | ATTUNet     | 34.88 M    | 66.54  | 2.08 | 269.80      |
> | AGNet| 9.33 M     | 16.73  | 5.99 | 330.72      |
> | ConvUNeXt   | 3.51 M     | 7.18   | 2.82 | 160.07      |
> | DCSAU-net  | 2.60 M     | 6.72   | 6.45 | 180.37      |
> | R2UNet      | 39.09 M    | 152.71 | 3.20 | 323.44      |
> | SAUNet      | 0.48 M     | 2.33   | 1.59 | 37.66|
> | BayeSeg     | 19.32 M    | 88.82  | 7.95 | 144.50      |
> | GraphSeg    | 19.32 M    | 88.82  | 7.78 | 145.95      |
>
>
> >> Q2. How sensitive is the method to the quality and representativeness of the statistical retinal atlas used for graph prior construction?
>
> A2. Thanks for your insightful question. In this work, the deformable graph prior provides a guidance for the Decomposition part to extract the graph-like structures for subsequent segmentation. As shown in Figure 4-6, the vascular shape $s$ extracted from the Decomposition is graph-like and the other parts, like color and shadow, are decomposed into the $m$. This is the source of our GraphSeg’s generalizability.
>
> The graph construction follows [1]. Graph Atlas is obtained by performing their code on the training set of CHASE. Following your suggestion, we also use the graph prior from the CHASE to train GraphSeg on the DRIVE dataset. The results are shown as follows: (1) GraphSeg(trained) outperforms UNet(trained) due to the graph prior, **which demonstrates that the graph prior extracted from one dataset can be transferred to the other dataset**. (2) GraphSeg outperforms UNet(trained) on most metrics, which confirms the generalizability of GraphSeg and demonstrates the robustness of our framework with respect to graph priors.
>
>
> - DRIVE (train)
>
> |Model|Acc|SoftDice|F1|Sensitivity|Specificity|
> |---|---|---|---|---|---|
> |GraphSeg(trained)|96.13|85.23|84.82|83.02|98.15|
> |Unet(trained)|90.89|61.09|61.09|48.55|98.51|
>
> - CHASE
>
> |Model|Acc|SoftDice|F1|Sensitivity|Specificity|
> |---|---|---|---|---|---|
> |GraphSeg|95.02|82.56|79.37|89.70|95.69|
> |Unet(trained)|95.17|79.74|78.39|81.97|96.79|
> |GraphSeg(trained)|96.54|82.91|81.82|87.84|97.40|
>
> - HRF
>
> |Model|Acc|SoftDice|F1|Sensitivity|Specificity|
> |---|----|---|---|----|---|
> |GraphSeg|93.91|78.20|77.25|78.40|96.34|
> |Unet(trained)|89.79|71.37|67.76|78.15|91.74|
> |GraphSeg(trained)|95.88|84.29|83.89|81.58|98.07|
>
> [1] PAMI’24. Statistical analysis of complex shape graphs.
>
> >>Q3. The paper appears to lack a comprehensive comparison with recent end-to-end deep learning approaches for vessel segmentation (e.g., U-Net variants, Transformer-based methods, recent CNN architectures).
>
> A3. Thanks for your suggestion. Following your suggestion, we compare GraphSeg with seven additional methods, including AttUNet [6], AGNet [7], ConvUNeXt [2], DCSAU-Net [3], R2UNet [4], SAUNet [8], and CDARL [9]. FRSGNet[1] is the current SOTA for retinal vessel segmentation.
>
> GraphSeg outperforms other methods in model generalization studies (on DRIVE and HRF), especially for F1 and Soft Dice. *For other methods, since the F1 scores and soft Dice scores are significantly lower on DRIVE and HRF, the acc, sensitivity, and specificity are not very meaningful due to the significant class imbalance between the foreground (vessels) and background*.
>
>
>
> - CHASE (train)
>
> | Model         | Acc   | Soft Dice | F1    | Sensitivity | Specificity |
> |---------------|-------|-----------|-------|-------------|-------------|
> | FSGNet        | 97.52 | 81.34     | 81.02 | 86.00       | 98.26       |
> | FRNet         | 97.26 | 80.10     | 79.10 | 84.74       | 97.65       |
> | AttUNet       | 97.54 | 66.02     | 77.06 | 78.92       | 98.56       |
> | AGNet         | 97.44 | 76.25     | 77.58 | 85.14       | 98.13       |
> | ConvUNeXt     | 97.21 | 72.40     | 73.93 | 75.97       | 98.39       |
> | DCSAU-net    | 97.17 | 74.33     | 74.68 | 79.87       | 98.12       |
> | R2UNet        | 96.96 | 67.99     | 71.73 | 74.60       | 98.20       |
> | SAUNet        | 97.47 | 76.17     | 77.23 | 82.02       | 98.33       |
> | CDARL (official) | 78.53 | 11.54   | 0.52  | 0.84        | 83.60       |
> | CDARL (Reproduce) | 83.18 | 16.97 | 15.69 | 26.05       | 86.93       |
> | BayeSeg       | 95.45 | 80.48     | 79.62 | 82.93       | 96.99       |
> | GraphSeg      | 96.54|82.91|81.82|87.84|97.40|
>
> - DRIVE
>
> | Model         | Acc   | Soft Dice | F1    | Sensitivity | Specificity |
> |---------------|-------|-----------|-------|-------------|-------------|
> | FSGNet        | 95.62 | 57.51     | 57.56 | 44.39       | 99.46       |
> | FRNet         | 96.65 | 71.33     | 73.57 | 67.53       | 98.85       |
> | AttUNet       | 95.07 | 45.27     | 50.22 | 38.75       | 99.30       |
> | AGNet         | 96.25 | 69.46     | 70.59 | 65.21       | 98.59       |
> | ConvUNeXt     | 97.07 | 61.91     | 63.77 | 74.35       | 97.92       |
> | DCSAU-net    | 95.30 | 62.30     | 62.56 | 56.82       | 98.19       |
> | R2UNet        | 95.01 | 63.86     | 67.02 | 70.14       | 96.60       |
> | SAUNet        | 96.46 | 69.78     | 70.67 | 61.92       | 99.06       |
> | BayeSeg       | 92.34 | 71.68     | 70.71 | 61.85       | 97.87       |
> | GraphSeg      | 93.56 | 77.16     | 76.01 | 68.09       | 98.19       |
>
> - HRF
>
> | Model         | Acc   | Soft Dice | F1    | Sensitivity | Specificity |
> |---------------|-------|-----------|-------|-------------|-------------|
> | FSGNet        | 95.90 | 68.26     | 68.56 | 71.50       | 98.28       |
> | FRNet         | 97.00 | 64.99     | 67.56 | 69.78       | 97.28       |
> | AttUNet       | 97.37 | 54.20     | 65.29 | 66.41       | 97.94       |
> | AGNet         | 96.70 | 63.03     | 64.54 | 66.69       | 97.10       |
> | ConvUNeXt     | 95.62 | 59.74     | 60.60 | 49.56       | 99.07       |
> | DCSAU-net    | 96.63 | 61.70     | 62.09 | 68.66       | 97.30       |
> | R2UNet        | 94.21 | 49.07     | 52.03 | 88.87       | 94.40       |
> | SAUNet        | 97.15 | 65.87     | 67.17 | 83.92       | 97.65       |
> | BayeSeg       | 92.05 | 70.73     | 70.13 | 67.71       | 96.13       |
> | GraphSeg      |93.38|78.16|76.59|78.90|96.88|
>
> [1] [ArXiv’25] FSG-Net: Full-scale Representation Guided Network for Retinal Vessel Segmentation
>
> [2] [Knowledge-based systems’22] ConvUNeXt: An efficient convolution neural network for medical image segmentation
>
> [3] [Computers in Biology and Medicine’23] DCSAU-Net: A deeper and more compact split-attention U-Net for medical image segmentation
>
> [4] [Journal of Computational Analysis & Applications’24] Recurrent residual convolutional neural network based on u-net (r2u-net) for medical image segmentation
>
> [5] [Biomedical Optics Express’24] FRD-Net: a full-resolution dilated convolution network for retinal vessel segmentation
>
> [6] [Medical Imaging with Deep Learning’18] Attention u-net: Learning where to look for the pancreas
>
> [7] [ISAIMS’20] AGNet：Retinal Blood Vessel Segmentation via Attention Gate Network
>
> [8] [MICCAI’20] Saunet: Shape attentive u-net for interpretable medical image segmentation
>
> [9] [MedIA'24] C-DARL: Contrastive diffusion adversarial representation learning for label-free blood vessel segmentation

---

### Official Review · Reviewer_Qe69 · 2025-07-03

**Clarity:** 2
**Significance:** 2
**Originality:** 3
**Rating:** 4
**Confidence:** 4

**Summary:**

The paper presents a learning-based Bayesian retinal vessel segmentation method to address classical Bayesian image segmentation models focusing on compact shapes. By proposing a deformable retinal graph prior, the proposed method can match heterogeneous vascular structures and achieve spatial alignment between the vessel branches and deformable graph priors. The Proposed method has been demonstrated on three retinal vessel datasets. While the results of the comparison study for each dataset show comparable performance between the proposed method and the baseline models, the generalization test results of the proposed method show superior performance over the baseline models.

**Questions:**

Please refer to the above weaknesses.

- It is uncertain whether the proposed method shows state-of-the-art performance when comparing the model to recent vessel segmentation methods.
- Please provide the computational cost comparison.
- Please perform a statistical test to see whether the performance of the proposed method is statistically different from the baseline models.

**Ethical Concerns:**

["NO or VERY MINOR ethics concerns only"]

**Final Justification:**

After reviewing the authors' rebuttal and the other reviewers' comments, I've increased my rating. The authors have addressed most of my concerns. The proposed method has been compared with recent self-supervised-learning-based methods and still showed superior performance to the existing methods. While the proposed method takes slightly longer time than the baseline models in the inference stage, the proposed method shows statistically higher performance, which would be helpful in vessel segmentation in real clinical practice.

**Limitations:**

yes.

**Paper Formatting Concerns:**

There is no concern.

**Quality:**

3

**Strengths And Weaknesses:**

### Strengths
- The proposed method handles one of the challenging problems of vessel segmentation in matching heterogeneous vessel structures.
- The proposed deformable graph prior effectively guides the image decomposition and the vessel segmentation through a variational Bayesian framework.
- The proposed method shows high generalizability performance when evaluating the model on different datasets from the training dataset.

### Weaknesses
- While there are many learning-based vessel segmentation methods that have been presented [1, 2, 3] recently and showed high generalization performance, the paper compares the proposed method to only three methods.
- The proposed method requires a higher computational cost to perform vascular structure matching in the inference stage, compared to the methods that only use neural networks.
- The performance improvement of the proposed method in the generalizability study is quite marginal.

[1] Li, Yang, et al. "Diffusion probabilistic learning with gate-fusion transformer and edge-frequency attention for retinal vessel segmentation." IEEE Transactions on Instrumentation and Measurement (2024).

[2] Kim, Boah, et al. "C-DARL: Contrastive diffusion adversarial representation learning for label-free blood vessel segmentation." Medical Image Analysis 91 (2024): 103022.

[3] Liu, Jianhua, et al. "HRD-Net: High resolution segmentation network with adaptive learning ability of retinal vessel features." Computers in Biology and Medicine 173 (2024): 108295.

---

> ### Author Rebuttal · Authors · 2025-07-30
>
> Thank you for your comments! All your concerns have been carefully addressed as below. The manuscript will be carefully revised accordingly. We sincerely hope our responses fully address your questions.
>
> >> Q1. It is uncertain whether the proposed method shows state-of-the-art performance when comparing the model to recent vessel segmentation methods.
>
> A1. Thanks for your suggestion. Following your suggestion, we compare GraphSeg with additional seven methods, including AttUNet [6], AGNet [7], ConvUNeXt [2], DCSAU-Net [3], R2UNet [4], SAUNet [8], and CDARL [9]. FRSGNet[1] is current SOTA for retinal vessel segmentation. We tried to compare with the suggested methods. However, the code for HRD-Net is not available, and the code for DPL-GTF-EFA cannot be executed properly. Finally, we provide two experimental results of CDARL. One was tested using the official checkpoint, the other was reproduced on CHASE.
>
> GraphSeg outperforms other methods in model generalization studies (on DRIVE and HRF), especially for F1 and Soft Dice. *For other methods, since the F1 scores and soft Dice scores are significantly lower on DRIVE and HRF, the acc, sensitivity and specificity are not very meaningful due to the significant class imbalance between the foreground (vessels) and background.*
>
>
> - CHASE (train)
>
> | Model         | Acc   | Soft Dice | F1    | Sensitivity | Specificity |
> |---------------|-------|-----------|-------|-------------|-------------|
> | FSGNet        | 97.52 | 81.34     | 81.02 | 86.00       | 98.26       |
> | FRNet         | 97.26 | 80.10     | 79.10 | 84.74       | 97.65       |
> | AttUNet       | 97.54 | 66.02     | 77.06 | 78.92       | 98.56       |
> | AGNet         | 97.44 | 76.25     | 77.58 | 85.14       | 98.13       |
> | ConvUNeXt     | 97.21 | 72.40     | 73.93 | 75.97       | 98.39       |
> | DCSAU-net    | 97.17 | 74.33     | 74.68 | 79.87       | 98.12       |
> | R2UNet        | 96.96 | 67.99     | 71.73 | 74.60       | 98.20       |
> | SAUNet        | 97.47 | 76.17     | 77.23 | 82.02       | 98.33       |
> | CDARL (official) | 78.53 | 11.54   | 0.52  | 0.84        | 83.60       |
> | CDARL (Reproduce) | 83.18 | 16.97 | 15.69 | 26.05       | 86.93       |
> | BayeSeg       | 95.45 | 80.48     | 79.62 | 82.93       | 96.99       |
> | GraphSeg      | 96.54|82.91|81.82|87.84|97.40|
>
> - DRIVE
>
> | Model         | Acc   | Soft Dice | F1    | Sensitivity | Specificity |
> |---------------|-------|-----------|-------|-------------|-------------|
> | FSGNet        | 95.62 | 57.51     | 57.56 | 44.39       | 99.46       |
> | FRNet         | 96.65 | 71.33     | 73.57 | 67.53       | 98.85       |
> | AttUNet       | 95.07 | 45.27     | 50.22 | 38.75       | 99.30       |
> | AGNet         | 96.25 | 69.46     | 70.59 | 65.21       | 98.59       |
> | ConvUNeXt     | 97.07 | 61.91     | 63.77 | 74.35       | 97.92       |
> | DCSAU-net    | 95.30 | 62.30     | 62.56 | 56.82       | 98.19       |
> | R2UNet        | 95.01 | 63.86     | 67.02 | 70.14       | 96.60       |
> | SAUNet        | 96.46 | 69.78     | 70.67 | 61.92       | 99.06       |
> | BayeSeg       | 92.34 | 71.68     | 70.71 | 61.85       | 97.87       |
> | GraphSeg      | 93.56 | 77.16     | 76.01 | 68.09       | 98.19       |
>
> - HRF
>
> | Model         | Acc   | Soft Dice | F1    | Sensitivity | Specificity |
> |---------------|-------|-----------|-------|-------------|-------------|
> | FSGNet        | 95.90 | 68.26     | 68.56 | 71.50       | 98.28       |
> | FRNet         | 97.00 | 64.99     | 67.56 | 69.78       | 97.28       |
> | AttUNet       | 97.37 | 54.20     | 65.29 | 66.41       | 97.94       |
> | AGNet         | 96.70 | 63.03     | 64.54 | 66.69       | 97.10       |
> | ConvUNeXt     | 95.62 | 59.74     | 60.60 | 49.56       | 99.07       |
> | DCSAU-net    | 96.63 | 61.70     | 62.09 | 68.66       | 97.30       |
> | R2UNet        | 94.21 | 49.07     | 52.03 | 88.87       | 94.40       |
> | SAUNet        | 97.15 | 65.87     | 67.17 | 83.92       | 97.65       |
> | BayeSeg       | 92.05 | 70.73     | 70.13 | 67.71       | 96.13       |
> | GraphSeg      | 93.38|78.16|76.59|78.90|96.88|
>
> [1] [ArXiv’25] FSG-Net: Full-scale Representation Guided Network for Retinal Vessel Segmentation
>
> [2] [Knowledge-based systems’22] ConvUNeXt: An efficient convolution neural network for medical image segmentation
>
> [3] [Computers in Biology and Medicine’23] DCSAU-Net: A deeper and more compact split-attention U-Net for medical image segmentation
>
> [4] [Journal of Computational Analysis & Applications’24] Recurrent residual convolutional neural network based on u-net (r2u-net) for medical image segmentation
>
> [5] [Biomedical Optics Express’24] FRD-Net: a full-resolution dilated convolution network for retinal vessel segmentation
>
> [6] [Medical Imaging with Deep Learning’18] Attention u-net: Learning where to look for the pancreas
>
> [7] [ISAIMS’20] AGNet：Retinal Blood Vessel Segmentation via Attention Gate Network
>
> [8] [MICCAI’20] Saunet: Shape attentive u-net for interpretable medical image segmentation
>
> [9] [MedIA'24] C-DARL: Contrastive diffusion adversarial representation learning for label-free blood vessel segmentation
>
>
> >> Q2. … requires a higher computational cost to perform vascular structure matching in the inference stage… Please provide the computational cost comparison.
>
> A2. Thanks for your suggestion, we provide the computational cost comparison as follows. The computational costs are comparable for GraphSeg to other methods. In the inference stage, we only need to run the Decomposition (green in Figure 2) and Segmentation (blue in Figure 2) parts. Therefore, the computational costs are comparable. Besides, without the Deformable graph prior, GraphSeg is downgraded to BayeSeg, thus the computational costs are almost the same for GraphSeg and BayeSeg for inference. We trained GraphSeg for 2.8 hours on GTX 4090. Since each method uses different training settings, it is not directly comparable. Therefore, we focus on inference metrics which are more relevant for practice.
>
> | Model       | Parameters | GFLOPs | Inference Time (s) | Memory (MB) |
> |-------------|------------|--------|--------------------|-------------|
> | FSGNet      | 18.32 M    | 89.59  | 6.23               | 925.42      |
> | FRNet       | 7.38 M     | 55.19  | 4.67               | 374.65      |
> | ATTUNet     | 34.88 M    | 66.54  | 2.08               | 269.80      |
> | AGNet       | 9.33 M     | 16.73  | 5.99               | 330.72      |
> | ConvUNeXt   | 3.51 M     | 7.18   | 2.82               | 160.07      |
> | DCSAU-net  | 2.60 M     | 6.72   | 6.45               | 180.37      |
> | R2UNet      | 39.09 M    | 152.71 | 3.20               | 323.44      |
> | SAUNet      | 0.48 M     | 2.33   | 1.59               | 37.66       |
> | BayeSeg     | 19.32 M    | 88.82  | 7.95               | 144.50      |
> | GraphSeg    | 19.32 M    | 88.82  | 7.78               | 145.95      |
>
> >> Q3. The performance improvement of the proposed method in the generalizability study is quite marginal. Please perform a statistical test to see whether the performance of the proposed method is statistically different from the baseline models.
>
> A3. Thanks for your comment. Following your suggestions, we perform a statistical test between GraphSeg and BayeSeg as follows. GraphSeg outperforms BayeSeg significantly, as shown in the table. For other methods, their F1 scores and soft Dice scores in the model generalization studies are lower than that of BayeSeg, and thus they also have significant difference with GraphSeg in generalizability.
>
> - CHASE DB
>
> | Metric      | t-test   | Wilcoxon | Significant (p < 0.05) |
> |-------------|----------|----------|------------------------|
> | Accuracy    | 0        | 0.00012  | TRUE                   |
> | Soft Dice   | 0.00001  | 0.00024  | TRUE                   |
> | F1          | 0        | 0.00012  | TRUE                   |
> | Sensitivity | 0.00078  | 0.00037  | TRUE                   |
> | Specificity | 0.00022  | 0.00061  | TRUE                   |
>
> - DRIVE
>
> | Metric      | t-test   | Wilcoxon | Significant (p < 0.05) |
> |-------------|----------|----------|------------------------|
> | Accuracy    | 0.00004  | 0.00021  | TRUE                   |
> | Soft Dice   | 0.00483  | 0.00639  | TRUE                   |
> | F1          | 0.00013  | 0.00059  | TRUE                   |
> | Sensitivity | 0.00554  | 0.00422  | TRUE                   |
> | Specificity | 0.00016  | 0.00059  | TRUE                   |
>
> - HRF
>
> | Metric      | t-test   | Wilcoxon | Significant (p < 0.05) |
> |-------------|----------|----------|------------------------|
> | Accuracy    | 0.00018  | 0.00002  | TRUE                   |
> | Soft Dice   | 0.00004  | 0        | TRUE                   |
> | F1          | 0        | 0        | TRUE                   |
> | Sensitivity | 0        | 0        | TRUE                   |
> | Specificity | 0.00502  | 0.00248  | TRUE                   |

---

### Decision · Program_Chairs · 2025-09-17

**Decision:**

Accept (poster)

**Comment:**

This paper presents an interesting formulation for retinal vessel segmentation, which integrates anatomical graph priors to enhance cross-dataset performance. During the review process, reviewers pointed to the lack of sufficient methods compared in the empirical validation, questioned the computational overhead introduced by the proposed approach, as well as the limited size of the employed datasets. After discussions, the authors successfully addressed most of these comments, which led to all the reviewers recommending the acceptance of this work, particularly due to its potential impact in cross-dataset settings. After reading all the comments, I believe that this work could be accepted, as it brings an interesting probabilistic segmentation method tailored to the retinal vessel segmentation task, and thus interesting for the medical community (or other domains whose images share structural similarities with vessels). Having said this, I also believe that additional datasets (e.g., AV-WIDE, IOSTAR, or UoA-DR, for example) could have been used to strengthen the empirical validation, as they are used in relevant papers and offer distinct challenges to the datasets employed (for example multi-centric and multi-devices images).